# Stratospheric Modulation of Arctic Oscillation Extremes as Represented by Extended-Range Ensemble Forecasts

Jonas Spaeth and Thomas Birner

Meteorological Institute Munich, Ludwig-Maximilians-University, Munich, Germany

**Correspondence:** Jonas Spaeth (jonas.spaeth@physik.uni-muenchen.de)

**Abstract.** The Arctic Oscillation (AO) describes a seesaw pattern of variations in atmospheric mass over the polar cap. It is by now well established that the AO pattern is in part determined by the state of the stratosphere. In particular, sudden stratospheric warmings (SSWs) are known to nudge the tropospheric circulation toward a more negative phase of the AO, which is associated with a more equatorward shifted jet and enhanced likelihood for blocking and cold air outbreaks in mid-latitudes. SSWs are also thought to contribute to the occurrence of extreme AO events. However, statistically robust results about such extremes are difficult to obtain from observations or meteorological (re-)analyses due to the limited sample size of SSW events in the observational record (roughly 6 SSWs per decade). Here we exploit a large set of extended-range ensemble forecasts within the subseasonal-to-seasonal (S2S) framework to obtain an improved characterization of the modulation of AO extremes due to stratosphere-troposphere coupling. Specifically, we greatly boost the sample size of stratospheric events by using potential SSWs (p-SSWs), i.e., SSWs that are predicted to occur in individual forecast ensemble members regardless of whether they actually occurred in the real atmosphere. For example, the S2S ensemble of the European Centre for Medium-Range Weather Forecasts gives us a total of 6101 p-SSW events for the period 1997-2021.

A standard lag-composite analysis around these p-SSWs validates our approach, i.e., the associated composite evolution of stratosphere-troposphere coupling matches the known evolution based on reanalyses data around real SSW events. Our statistical analyses further reveal that following p-SSWs, relative to climatology: 1) persistently negative AO states ($> 1$ week duration) are 16% more likely, 2) the likelihood for extremely negative AO states ($< -3\sigma$) is enhanced by about 40-80%, while that for extremely positive AO states ($> +3\sigma$) is reduced to almost zero, 3) approximately 50% of extremely negative AO states that follow SSWs may be attributable to the SSW, whereas about one quarter of all extremely negative AO states during winter may be attributable to SSWs. A corresponding analysis relative to strong stratospheric vortex events reveals similar insights into the stratospheric modulation of positive AO extremes. However, conclusions in terms of causality remain difficult, in part due to unconsidered confounding factors.

## 1 Introduction

Day-to-day variability of the northern extratropical hemispheric-scale circulation during winter is dominated by the so-called Northern Annular Mode (NAM, Thompson and Wallace, 1998). The surface manifestation of the NAM is often referred to as Arctic Oscillation (AO). This variability pattern primarily describes fluctuations of atmospheric mass over the polar cap

with associated opposite fluctuations on its equatorward flank. In its positive phase the AO corresponds to decreased mass over the polar cap with associated strengthened pressure gradient across mid-latitudes that goes along with a stronger polar-front/eddy-driven jet that is shifted poleward and more zonally aligned. Likewise, in its negative phase the jet is weakened, shifted equatorward and often more meriodonally distorted.

Although a single index cannot represent the entire extratropical weather, it indicates tendencies towards certain weather patterns, which in turn can also have strong local effects. Especially AO values that deviate considerably from 0 (the climatological mean) are rare, by construction, and can often be associated with strong *local* weather extremes (Thompson and Wallace, 2001): For instance, the daily AO index was around $-2.5$ in winter 2009/10, which was accompanied by record cold snaps and snow fall over large parts of the United States, Europe and East Asia (Cohen et al., 2010). In winter 2019/20, extreme

storminess over Central Europe occurred during a highly positive AO phase with wind gusts of up to 177 km/h being recorded over Germany (Haeseler et al., 2020). Furthermore, Kim et al. (2020) report increased likelihood of Siberian wildfires in April following positive AO periods in February and March.

The AO can also be influenced by "external" weather patterns and one prominent teleconnection exists between the AO and the stratospheric polar vortex. The latter describes a strong westerly wind band around 60°N extending over 10 hPa, which

forms every year in winter (Waugh et al., 2017). Numerous studies show that, on average, a very strong polar vortex (SPV) is associated with a strengthened circumpolar flow in the troposphere - as indicated by a positive AO index (e.g., Baldwin and Dunkerton, 2001; Lawrence et al., 2020; Rupp et al., 2021). The reverse is true for a weak polar vortex, with such events being a special case: The breaking of planetary waves in the stratosphere and the associated westward forcing can lead to a complete breakdown of the polar vortex. In these cases, the zonal mean zonal wind reverses and the climatologically dominant

westerly winds are replaced by weak or moderate easterlies. During the vortex disruption, air masses converge in the center of the vortex and are forced to sink. The accompanying strong and rapid adiabatic heating is the reason that such extreme weak vortex events are called sudden stratospheric warmings (SSWs, Baldwin et al., 2021). SSWs are observed about 6 times per decade and are, as described previously, associated with a negative AO index on average. On synoptic scales, SSWs have also been tight to subsequently favored occurrence of certain weather regimes over the North Atlantic (Domeisen et al., 2020b) and

over North America (Lee et al., 2019).

Consistent with the local implications of a negative AO index, SSWs can for example lead to cold spells in Northern Europe and increased storminess over Southern Europe (Domeisen and Butler, 2020b, and references herein). Whether it is generally valid that SSWs and also strong polar vortex events lead to a subsequently more likely occurrence of AO extremes (and associated local extremes) is difficult to analyze because the statistical links are weak in each case, i.e., not each SSW/SPV

event is followed by an AO extreme. Therefore, a very large sample of SSW and SPV events are needed to quantify the subsequent risk increase of AO extremes. However, reanalyses data only cover about 40-70 years, depending on the data set, and thus about 30-40 SSWs - too few to robustly determine conditional probabilities (e.g., given a stratospheric extreme event, how likely is a following tropospheric extreme event).

In order to allow for analyses of larger event sample sizes, past studies have used, for example, idealized model simulations

(e.g., Hitchcock and Simpson, 2014; Jucker, 2016). Even though such models have proven to be useful to develop a qualitative

and conceptual picture, they often show a weaker tropospheric response to stratospheric events compared to observational data (Gerber et al., 2009). In this study, we aim to improve the characterization of coupled stratospheric and tropospheric circulation extremes using operational, state-of-the-art, extended-range forecasts. Relatively large ensembles, frequent model initializations and the generation of hindcasts allow us to analyze a large set of predicted SSWs and SPV events (p-SSWs/
p-SPVs, see discussion in section 2). Although the vast majority of these p-SSWs did not materialize in the real atmosphere we show that they nevertheless provide reliable statistical information about stratosphere-troposphere coupling. Our analyses implicitly assume that each ensemble member corresponds to a possible real-atmospheric evolution. The diagnosed p-SSWs include false alarm events (see, e.g., Taguchi, 2020), which we assume are based on the same underlying physics as those SSWs that occurred in the real atmosphere. Furthermore, the individual evolution (related to forecast score) is arguably not
relevant for statistical characterizations of circulation anomalies.

The analysis is thus based on the assumption that the forecast models simulate the observed variability of the AO sufficiently well, including its modulation due to stratospheric variability. High-top models, in particular, show realistic stratosphere-troposphere coupling (Domeisen and Butler, 2020a; Domeisen et al., 2020a). However, due to the small sample size of observed events, it is generally difficult to conclude whether any discrepancies between model and observational data are due to model
or sampling errors. For this study, we will show that the models agree with observations in established diagnostics that can be robustly derived from reanalyses, including, e.g., the frequency of SSWs, their seasonality and their average impact on the subsequent AO. Although our quantitative statistical analyses cannot be compared directly to observational data, they may be considered as best estimate given the currently available observational record and modeling capabilities.

We will compute statistical measures that combine conditional and baserate probabilities for stratospheric and AO extreme
(co-)occurrences and in order to address our following research questions:

1. By how much is the probability of persistently positive or negative AO phases increased following stratospheric polar vortex extremes?

2. By how much is the probability of subsequent AO extremes increased following stratospheric polar vortex extremes?

3. What fraction of AO extremes may be attributable to preceding stratospheric polar vortex extremes?

To illustrate which AO extremes are classified as "attributable", consider the following scenarios where a stratospheric event is followed by an AO extreme: In relation to the AO extreme the stratospheric extreme may

(*a*) represent a necessary and sufficient cause

(*b*) represent one among multiple contributory causes

(*c*) be caused by a confounding factor, which also causes the AO extreme

(*d*) not be causal

In scenario *a*, the AO extreme is attributable to the preceding stratospheric event, whereas it is not attributable in scenario *d*. In scenario *b*, disentanglement of different contributory factors is difficult. Each involved process can but does not need to be also a necessary cause. (Consider for example a situation where an AO extreme would have occurred also without a preceding stratospheric extreme, but the stratospheric extreme resulted in a stronger or earlier manifestation.) In this study, we do

neither aim to disentangle the multiple involved pathways *a-c* nor to provide a rigorous quantification of causality (which is itself ambiguous in a complex system). Instead, we estimate how many AO extremes may be attributable to the stratospheric extreme, which refers to the fraction that would have statistically not occurred without the stratospheric event. Importantly, scenario *c* shows that "without the stratospheric event" requires to also remove any confounding factors. The analysis follows an observational approach (which is based on post-hoc computation of conditional probabilities) rather than a counterfactual approach (which is based on active interventions in the system; Pearl, 2009, see section 8 for a more detailed interpretation of the results with respect to causality). However, even without disentangling scenarios *a*, *b* and *c*, the observational approach provides relevant and practical insights into the statistical association between and the importance of stratospheric and subsequent AO extremes.

The paper is organized as follows: Section 2 provides an overview of the extended-range forecasts used in this study. Section 3 defines stratospheric and tropospheric circulation extremes and presents basic event statistics. For SSWs, we validate in section 4 that the predicted events agree, in well-known diagnostics, with events that are identified in reanalysis data. This motivates section 5, where the probability of AO extremes following predicted SSWs is analyzed. Conversely, section 6 shows how often predicted AO extremes are preceded by predicted SSWs and how many AO extremes may be attributable to preceding SSWs. Section 7 reveals in a similar fashion the statistical relation between predicted strong polar vortex events and predicted positive AO extremes, before the key findings are discussed and summarized in section 8.

## 2 Description of extended-range ensemble forecasts

The subseasonal to seasonal (S2S) prediction project (Vitart et al., 2017) provides a collection of extended-range (up to 60 days lead time) ensemble forecasts from different weather services. Forecasts differ in terms of model specifications (e.g., spatial resolution, parameterizations, maximum lead time). All forecast systems create hindcasts in addition to the realtime forecasts in order to calibrate the forecasts and to allow the construction of the model's climatology. For our application, the most relevant demand is an accurate representation of the stratosphere and in particular of stratosphere-troposphere coupling. Furthermore, a forecast model with a large number of hindcasts is beneficial, because it allows for more robust analyses by including multiple past years. Lastly, a large maximum lead time is needed as we want to identify extreme events in the forecasts and are then also interested in the time periods before and after the event.

We choose to use ECMWF and UKMO forecasts for this study, as these models best meet the above listed requirements. Importantly, both models have been demonstrated in previous studies to have a realistic representation of stratosphere-troposphere coupling (Domeisen and Butler, 2020a; Domeisen et al., 2020a).

For the decision on which initialization dates to use for the analyses, a trade-off has to be made between having as large a sample as possible and the fact that the forecast models are updated about every 1-3 years. Since late 2016, the ECMWF model (CY43R1) has been running at a higher horizontal resolution. Therefore, to avoid mixing forecasts before and after 2016, forecasts from winter 2017/18 up to and including 2020/21 are analyzed. Note that a minor model version change occurred in 2019, where initial conditions for the hindcasts are then obtained from ERA5 instead of ERA-Interim. However, we do

**Table 1.** Dataset specifications.

|  | S2S ECMWF | S2S UKMO | ERA5 |
|---|---|---|---|
| Type | Forecast | Forecast | Reanalysis |
| Vertical Res. | L91 | L85 | L137 |
| Time Range | d 0-46 | d 0-60 | 1979-2021 |
| Realtime | 51 member, 2 inits / week | 4 member, daily inits | - |
| Hindcast | 11 member, 2 inits / week, past 20y | 7 member, 4 inits / month, 1993-2015 | - |
| # Realtime Ens. Used | 114 | 396 | - |
| # Hindcast Ens. Used | 2280 | 1173 | - |
| # Individual Model Runs | 30894 | 9795 | - |

not expect this to be a major limitation for our analyses, as we are mostly interested in the overall statistical behavior of stratosphere-troposphere coupling, as opposed to single forecast performance.

We focus on Northern winter dynamics by analyzing forecasts initialized between mid-November (11/16) and end of February (02/22). For the four winter seasons, the ECMWF model thus features 114 real-time ensemble forecasts of 51 members each and 2 280 ensemble hindcasts of 11 members each. This results in a total of 30 894 individual model runs, all of which we refer to as "forecasts" for simplicity. For consistency, UKMO forecasts are used from the same initialization period, leading to 9 795 forecasts available for this model. A summary of the key specifications of the forecasts is given in Table 1, along with details of the ERA5 data (Hersbach et al., 2020) used.

## 3 Event statistics of stratospheric and tropospheric circulation extremes

### 3.1 Data sets and overall methodology

Each of the forecasts from the total set of 30 894 ECMWF forecasts provides a 47-day time series of the evolution of the atmosphere (UKMO: 61 days). In this study, we define specific events and then scan each forecast for the occurrence of such an event. If there are multiple events of one event type within one forecast, only the first event is used. Note that, by definition, all identified events are predicted events, but each may or may not actually occur in the real atmosphere. To highlight this aspect, and to avoid confusion with actual real-atmospheric events, the events identified in the forecasts may be denoted with a "p"-prefix, where "p" stands for "predicted" (alternatively, it could be thought of as "potential" for some aspects). In this study, all event composites and computed probabilities refer to predicted events.

For both datasets, ECMWF and UKMO, all individual forecast runs are treated equally and independently. This assumption is violated especially for forecasts belonging to the same ensemble. In fact, at initialization time these forecasts agree *entirely* except for ensemble perturbations. The individual members diverge from each other only with increasing lead time, when the predictability of the atmospheric flow gradually decreases. For this reason, we analyze only those events that occur at or after a forecast lead time of 10 days. It is assumed that initial condition-memory has sufficiently reduced by this point so that no two

individual forecasts fully match, and the same is thus true for the evolution of the identified events. This ensures a degree of statistical independence. The use of hindcasts further guarantees sampling of different boundary conditions, such as due to the El-Niño-Southern-Oscillation, the Maddan-Julian-Osciallation or sea ice variations.

Furthermore, it is ensured that for each identified event both negative and positive lags can be considered. Due to the finite maximum lead time of each forecast, this demand is generally limited. For a predicted event that occurs early in the forecast (but after 10 days at the earliest), only a short period before the event can be examined, and the reverse is true for an event that occurs shortly before the end of the forecast. Therefore, to ensure a minimum common lag time that can be analyzed, events are additionally required to occur no later than 10 days before the end of the forecast. Consequently, events are allowed to occur between day 10 and 36 for ECMWF forecasts and between day 10 and 50 for UKMO forecasts. Thus, for all events, the lag period ±10 days can be examined, but with increasingly longer positive and negative lag times, fewer and fewer events contribute to the composite.

Extreme events are defined that refer to exceptional anomalies in the stratospheric and tropospheric circulation, respectively. As a measure of the strength of the stratospheric polar vortex we use the zonally averaged zonal wind at 10hPa at 60°N, hereafter referred to as u60.

## 3.2 Predicted SSWs

We define Sudden Stratospheric Warmings (p-SSWs), as days when u60 transitions from positive to negative, i.e., the polar vortex breaks down. As explained above, we do not include p-SSWs predicted within the first 10 days after forecast initialization. However, for p-SSWs, u60 is required to be solely positive within these first 10 days to ensure an intact westerly polar vortex at the start of the forecast. Following this event definition, we identify 6 101 p-SSWs in the ECMWF and 2 716 p-SSWs in the UKMO model.

Moreover, the analyses were repeated with a modified event definition, which we call *dynamical SSWs*, in order to investigate potential sensitivities. Dynamical SSWs were defined as a subset of SSWs, where in addition to the sign change, u60 is required to drop at least 20 ms$^{-1}$ averaged over $-5$ to $+5$ days lag relative to the SSW central date. Thereby, this event definition forms the intersection between SSWs (following Charlton and Polvani, 2007) and sudden stratospheric deceleration events (following Birner and Albers, 2017, ensuring a rapid deceleration around the event central date). Our results reveal only modest quantitative differences between SSWs and dynamical SSWs and we therefore focus on SSWs only, to allow better comparison with other studies.

In Figure 1 we provide an overview about the distribution of ECMWF p-SSWs as a function of the year, forecast lead time and calendar month (see Fig. S1 for a corresponding analysis of UKMO forecasts). p-SSWs are found for all winter seasons considered. Absolute numbers are presented to show which winter seasons contribute how many events to the analysis. Due to the realtime-hindcast-setup, the number of underlying forecasts varies across winter seasons. Therefore, we additionally provide a proxy for the SSW probability per winter season to illustrate inter-annual variability (see appendix B for details).

The largest number of events is identified in the winter season 2017/18, which includes also the most forecasts (realtime 2017/18 plus hindcasts related to initializations from 2018/19 to 2020/21). Different factors lead to a highly varying number

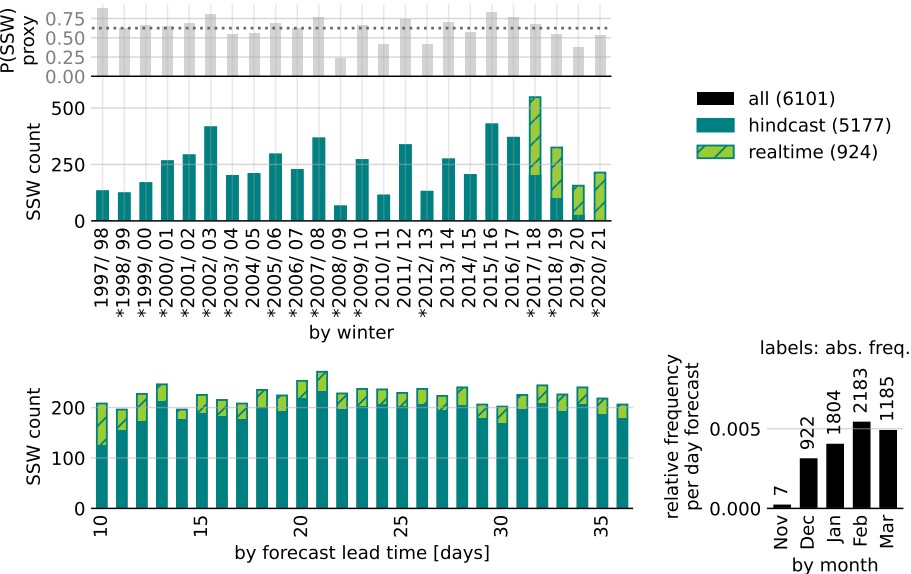

**Figure 1.** Distribution of analyzed p-SSWs in ECMWF forecasts. Absolute event counts (center left) and seasonal probability proxy (top left), grouped by winter. Asterisks denote years with real atmosphere SSWs (Butler et al., 2017). Grouped by forecast lead time (bottom left) and by month (bottom right).

of events between the different years. These include internal dynamic variability, a slightly varying number of underlying

forecasts, due to the realtime/ hindcast prediction setup, and the varying number of events per winter due to the evolution of the polar vortex of the real atmosphere in the respective winter, which determines the initial conditions of the forecasts.

A forecast that is initialized with a strong polar vortex tends to maintain a strong polar vortex and produces fewer SSWs compared to a forecast with an initially weak polar vortex. Moreover, forecasts that do not start with ten consecutive days of positive u60 are discarded by default. Thus, if the polar vortex in the real atmosphere is already easterly at the initialization time

or is predicted to become easterly within the first ten days, such forecasts will not contribute any events to the analysis. This can be illustrated by the example of the 2009th SSW (24 January 2009, see Butler et al., 2017). The event had low predictability at lead times longer than 8 days (Karpechko, 2018). Before the event, between end of December 2008 and mid-January 2009, the polar vortex was exceptionally strong, leading to an only marginal SSW probability in the forecasts and suggesting that the event itself was unlikely given the prevailing dynamics[1]. As a result, 2008/09 shows the lowest number of SSWs: In the first

winter half up to initialization dates around mid-January, hardly any events were predicted due to the relatively strong polar vortex. Later, forecasts predicting the real atmosphere SSW only did so at less than +10 day lead time, such that those events are discarded. Later initializations up to mid-February are excluded, because these do not predict persistent positive u60 within

---

[1]This also seems consistent with the interpretation of this event to fall under the category of self-induced resonance, which requires conditions (vortex geometry etc.) to be "just right" (see discussion in Albers and Birner, 2014).

the first 10 days lead time, due to the preceding SSW. As a result, winter season 2008/ 09 contributes only 64 (UKMO: 22) p-SSWs to the analysis, and at 23% (UKMO: 41%), the approximated SSW probability is the smallest in the period considered.

Based on the average number of 226 events per day lead time in the ECMWF model (cf. bottom left panel in Fig. 1), we estimate the probability of a SSW between mid-November and end of March, which yields 63% (see appendix B for details). This is consistent with the number of observed SSWs in reanalyses that is roughly 6 per decade (Butler et al., 2015).

While the rate of events per forecast day fluctuates only weakly in the ECMWF model, it moderately increases with lead time in the UKMO model (Fig. S1, bottom left panel). One might expect this to be due to the longer maximum lead time of the UKMO model (+60 days) compared to the ECMWF model (+46 days), which may allow more final-warming-like events. However, we find that the trend is still apparent when all forecasts initialized in February are excluded from the analyses (not shown).

Consistent with reanalyses (e.g, Ayarzagüena et al., 2019) and across both, the ECMWF and the UKMO model, the p-SSW frequency shows a maximum in February (bottom right panel in Fig. 1). However, Lawrence et al. (2022) find leadtime-dependent inconsistencies in the seasonal distribution of SSW probability compared to the observational record.

### 3.3 Predicted strong vortex events

Past literature has identified stratosphere-troposphere coupling not only following SSWs, but also following strong polar vortex events (SPVs, e.g., Baldwin and Dunkerton, 2001). However, the definition of a single event in these cases is somewhat more ambiguous, as there is no dynamically motivated threshold for u60, compared to 0 ms$^{-1}$ for SSWs. In addition, the dynamical changes in cases of a strong polar vortex are generally less abrupt, making it harder to pin down one particular central event day. For these reasons, we focus mainly on SSWs in this paper, however, we also provide a summary of the key results for SPV analyses in section 7. In these analyses, p-SPVs are defined as the first day on which u60 exceeds a threshold that, based on percentiles, represents the "opposite" of the SSW threshold of 0 ms$^{-1}$. Depending on the model's climatology, this threshold describes approximately the 91st percentile of the u60 distribution and is around 47 ms$^{-1}$.

### 3.4 Predicted AO events

In the troposphere, we define extreme events based on the Arctic Oscillation Index (short: AO; equivalent to the Northern Annular Mode Index at 1000 hPa, short: NAM1000). The index is calculated by first area-weighting the geopotential field between 65 and 90°N by the cosine of latitude and then averaging over the entire polar cap. The AO index then is the negative standardized anomaly of the obtained quantity. For technical details about the deseasonalization via the hindcasts, the reader is referred to appendix A. The positive phase of the AO describes a negative geopotential anomaly over the polar cap and a thereby induced enhanced circumpolar westerly circulation. Conversely, a negative AO reflects a weaker westerly circulation, which is typically associated with a southward shift of the jet that is also zonally more distorted.

We define tropospheric extreme events as the first day when the AO falls below a certain negative threshold (e.g., AO$^{-3}$ corresponds to AO $< -3$) or exceeds a certain positive threshold (e.g., AO$^{+3}$ corresponds to AO $> +3$). After testing different

thresholds, we opt for thresholds of up to 3 standard deviations which represents a tradeoff between severity of event and sufficiently large resulting sample sizes.

### 3.5 Conditional probabilities of polar vortex and AO extremes

In this study, conditional probabilities are computed to estimate the modulated likelihood of AO extremes under the presence or absence of preceding stratospheric extremes. For example, we expect the probability of at least one $AO^-$ extreme during
a given time period to be higher if that time period follows a SSW compared to the case that it does not follow a SSW. This is somewhat akin to the situation in climate attribution science, where one aims to quantify the increased risk of an extreme event due to anthropogenic climate change (e.g., National Academies of Sciences, Engineering and Medicine, 2016), or to the situation in epidemiology, where one aims to quantify the increased risk of contracting a disease given an exposure to a particular factor (e.g., smoking in the case of lung cancer; Peto, 2000). In such situations one may quantify the additional risk
due to the exposure based on the so-called relative risk increase (RRI):

$$\text{RRI} = \frac{\text{risk among the exposed}}{\text{risk among the unexposed}} - 1$$

In climate attribution science "exposure" may be thought of as "under the influence of anthropogenic climate change", whereas lack of exposure (the condition in the denominator) may be thought of as "without the influence of climate change" (e.g., based on pre-industrial control climate). In our case of stratosphere-troposphere coupling exposure may be thought of as "given that
a stratospheric extreme occurred". However, lack of exposure has to be evaluated with care. For example, assume that a given day $t_0$ fulfills the condition of "no stratospheric extreme" and an AO extreme occurs within a given period following $t_0$. This AO extreme cannot necessarily be considered "unexposed" as a stratospheric extreme may have occurred between $t_0$ and the date of the AO extreme. For our analyses that evaluate the increased probability of an AO extreme following a stratospheric extreme event we therefore adopt a modified version of RRI, where we replace the denominator with the risk of AO extreme
occurrence for the population (i.e., including both exposed and unexposed). To avoid confusion we will refer to this modified RRI simply as "relative probability increase" (RPI, see section 5). A negative RPI indicates that AO extremes become less likely following stratospheric events. The more positive the RPI, the more likely subsequent AO extremes become and the better does the stratospheric event serve as predictor for AO extremes.

One way to circumvent the above discussed issue of conditioning onto "unexposed" is to swap the conditioning. That is, we
may condition onto the occurrence of an AO extreme and evaluate the probability that a given preceding time period showed at least one day with stratospheric extreme occurrence – in this case the AO extreme is considered to be "exposed". Likewise, if the preceding time period shows no occurrence of stratospheric extreme, the AO extreme is considered to be "unexposed". Using Bayes theorem this allows us to estimate the fraction of attributable risk (FAR) of AO extremes to a preceding stratospheric extremes. FAR quantifies the reduction in the fraction (0 to 1) of AO extremes without preceding stratospheric events (and
without any confounding factors, see discussion in section 8). We will distinguish FAR among the exposed and among the population (see section 6).

**Table 2.** Definitions for (conditional) predicted SSW and AO events. Subscript $wt$ is short for "within time $t$". AO events can be negative ($AO^-$) or positive ($AO^+$) and may refer to a prescribed threshold, i.e., $AO_{wt}^{-3}$ or $AO_{wt}^{+3}$ correspond to "at least one day below $-3$ or above $+3$ within time $t$".

| Event | Description |
|---|---|
| $AO$ | probability that any day shows an AO extreme |
| $AO_{wt}$ | probability that any period of time $t$ shows at least one AO extreme |
| $AO_{wt} \mid SSW$ | given a SSW, probability of at least one AO extreme within subsequent time $t$ |
| $SSW_{wt}$ | probability that any period of time $t$ shows at least one SSW event |
| $\neg SSW_{wt}$ | probability that any period of time $t$ shows no SSW event |
| $SSW_{wt} \mid AO$ | given an AO extreme, probability of at least one day with u60 $< 0$ within preceding period of time $t$ |
| $\neg SSW_{wt} \mid AO$ | given an AO extreme, probability of no day with u60 $< 0$ within preceding period of time $t$ |
| $AO \mid SSW_{wt}$ | given a preceding period of time $t$ with at least one day with u60$< 0$, probability of AO extreme on day afterwards |
| $AO \mid \neg SSW_{wt}$ | given a preceding period of time $t$ with no day with u60$< 0$, probability of AO extreme on day afterwards |

Relative probability increase, attributable risk among the exposed and among the population all quantify, from different perspectives, the increased likelihood of AO extremes following stratospheric events. Mathematical definitions of how they are derived from baserate and conditional probabilities are introduced in the respective sections. We here provide an overview table about the event definitions that will be used (Tab. 2).

## 4 Evaluation of stratosphere-troposphere coupling based on predicted SSWs

To provide a baseline for our more detailed statistical analyses in the following sections, we first evaluate the general behavior of stratosphere-troposphere coupling based on p-SSW events in the S2S data. To do so we focus on the lag-composite evolution of the AO index relative to p-SSWs compared to real-atmospheric SSWs from ERA5. In addition, we show the NAM index at 200hPa (short: NAM200) because the lower stratosphere has been found to play an important role in stratosphere-troposphere coupling (e.g, Karpechko et al., 2017; White et al., 2020).

Figure 2 shows the evolution of u60 (top), NAM200 (center) and AO (bottom) during SSWs, averaged over all events, separately for ECMWF and UKMO. In addition to the composite mean, the 33rd to 66th percentiles across all ECMWF events on the respective lag day are shown. By construction, 100% of all events (ECMWF: 6 101, UKMO: 2 716) contribute to lag days $\pm 10$. For larger positive or negative lags, some forecasts have reached their maximum forecast lead time or have not yet been initialized. Therefore, the number of events drops off, which makes the statistics less robust: For the ECMWF model, the number of contributing events falls below 20% for lags smaller than $-31$ and larger than +31 days (UKMO: smaller than $-44$ and larger than +39 days).

By construction, u60 transitions from westerly to easterly at lag 0. Anomalies of u60 are slightly positive ahead of $-14$ days lag, which we interpret as an indication for vortex preconditioning (McIntyre, 1982; Albers and Birner, 2014; Jucker and

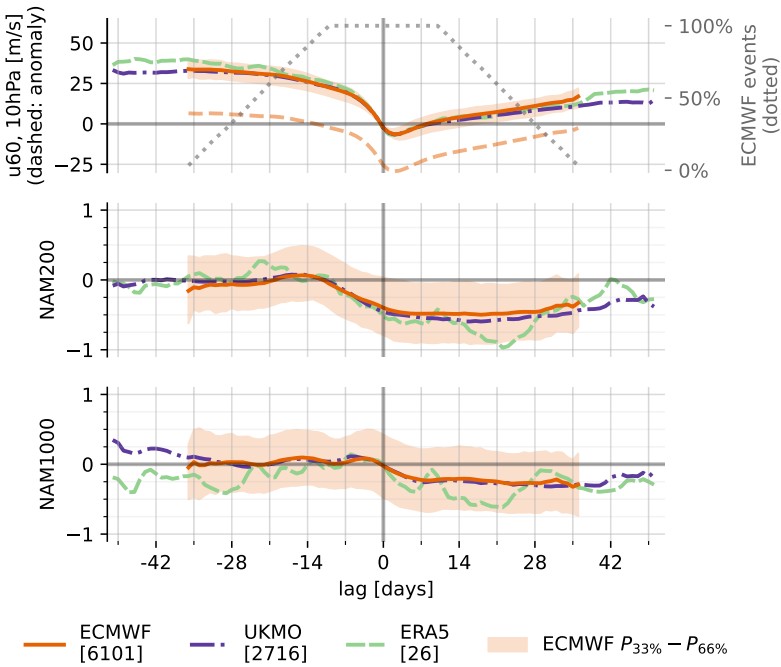

**Figure 2.** Lagged composite evolution of u60 (top panel), NAM200 (middle panel) and NAM1000 (=AO, bottom panel) relative to p-SSWs (ECMWF, UKMO) and SSWs (ERA5). It is presented the mean across all ECMWF events (orange, solid), the 33rd to 66th percentiles across all ECMWF events (orange, shaded), the mean of all UKMO events (purple, dash-dotted) and the mean across all ERA5 events (green, dashed). In the top panel further denoted are the average u60 anomalies (orange, dashed) and the relative number of contributing events to the composite in the ECMWF model (gray, dotted). Square brackets denote the total number of events, for each dataset.

Reichler, 2018). The anomalies become negative within the second week prior to the event central date. The largest average negative anomalies occur only few days after the event central day (lag +2 days: $-6\text{ms}^{-1}$). Afterwards, the vortex reestablishes and the average anomalies reach zero again after approximately 35 days. Consistent with, e.g., Baldwin and Dunkerton (2001), both NAM200 and AO are negative following the event. The shift of the NAM200 happens earlier (at lag day $-11$) and the

timing aligns well with the weakening of the polar vortex at 10hPa. The NAM200 anomaly is also more pronounced ($\approx -0.5$) compared to the AO ($\approx -0.3$). Interestingly, the AO distribution is slightly shifted toward positive values in the week prior to the central date, which is also robust for other diagnostics like the 10th, 30th, 70th and 90th percentiles (not shown). At long positive lag times, the NAM indices at 200 and 1000hPa are still negative (ECMWF: lag +36 days, UKMO: lag +51 days), but the trend goes to weaker negative values again.

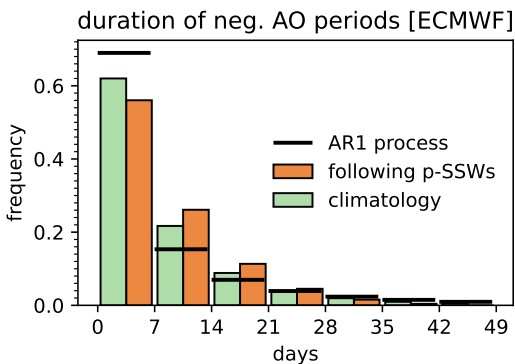

**Figure 3.** Histogram of the duration of negative AO periods, quantified by the number of consecutive days of AO < 0 and binned by 7-day chunks. Periods following ECMWF p-SSWs (orange bars, right half) are compared to the ECMWF model's climatology (green bars, left half) and a random first order auto-regressive model of the same 1-day-lag-autocorrelation as the AO in ERA5 (black, horizontal lines). ERA5 climatology is not shown, but agrees very well with the ECMWF forecast climatology.

Overall, the results are in agreement with ERA5 and previous literature and especially the evolution of u60 is remarkably similar. The negative NAM response at 200hPa and 1000hPa seems to be slightly stronger in the reanalysis, however, it is also noisier due to the smaller sample size.

## 5 Predicted AO extremes following predicted SSWs

In the following, we will exploit the larger available sample size of p-SSW events to diagnose and estimate whether the shift of the average AO index towards negative values is caused by 1) more persistent negative AO phases and/or 2) an increased probability of AO$^-$ extremes.

### 5.1 Persistence of negative AO phases

Figure 3 presents a histogram of the duration of predicted negative AO phases in the ECMWF model, binned into 7 day chunks. The duration is defined as the number of consecutive days with negative AO. The climatology serves as a reference including all 30 894 ECMWF forecasts used for this study. With approximately 62%, most phases of negative AO are shorter than 8 days. As another reference, a first order autoregressive model was set up with zero mean and standard deviation of 1, which may serve as a baseline. Its 1-day-autocorrelation is chosen to match the ERA5 AO timeseries and for robustness, it is estimated by averaging the lag-1-autocorrelation and the square-root of the lag-2-autocorrelation, yielding 0.91. ECWMF (S2S) and ERA5 agree very well in terms of climatology and lag-1-autocorrelation (not shown). However, the AO climatology shows short negative phases ($\leq 7$ days) less often and long negatives phases ($\geq 8$ days) more often compared to the AR1 process, indicating an AR1 process cannot reproduce AO variability.

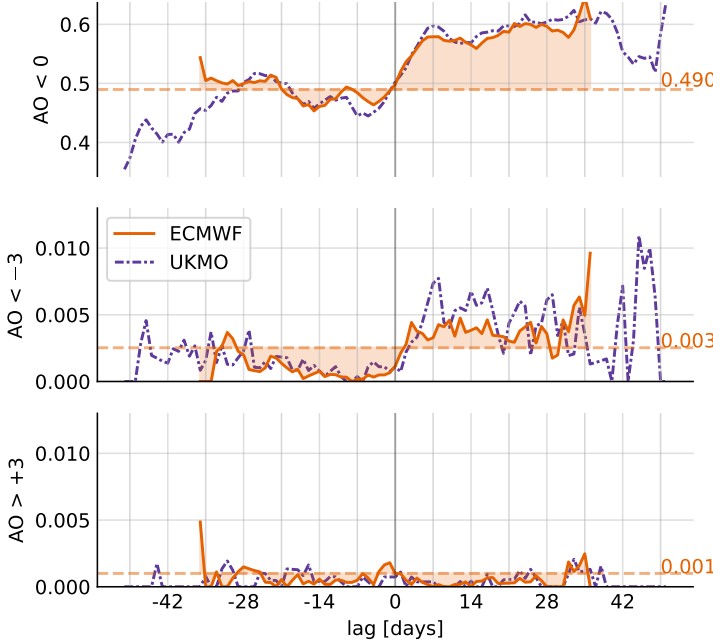

**Figure 4.** Daily probabilities of AO<0 (top panel), AO< −3 (middle panel) and AO>+3 (bottom panel) relative to p-SSWs, quantified by the fraction of events fulfilling the respective condition, separately for ECMWF (orange, solid) and UKMO (purple, dash-dotted). Day 0 corresponds to the p-SSW central date. In addition, probabilities are compared to the corresponding daily ECMWF climatology (dashed horizontal lines).

In addition, the diagnostic is presented for periods following p-SSWs. Here, the AO index is analyzed between lag day +1 relative to the event date and the maximum available lag time, which ranges between +10 and +36 days, depending on the forecast lead time when the event happens. Similar to the reference climatology, this diagnostic also underestimates the occurrence of long negative AO periods as the forecasts have finite maximum lead time. Nevertheless, periods following SSWs show a reduced frequency of shorter and an increased frequency of longer negative AO periods, compared to the climatology (and thus also to the AR1 process): For instance, 38% of negative AO periods are longer than 7 days in the climatology, whereas this probability rises to 44% following p-SSWs, which corresponds to a relative increase of 16%.

Sampling uncertainties turn out to be negligible within 95% confidence intervals. A similar analysis based on UKMO data shows very good quantitative agreement (not shown), which further confirms the robustness of the results.

### 5.2 Modulated probability of AO extremes

It is known that SSWs shift the subsequent AO distribution (see Fig. 2). This also implies an increased daily probability of negative and a reduced probability of positive AO extremes compared to their respective climatological probabilities. Fig. 4 shows the probabilities of negative ($< 0$), extremely negative ($< −3$) and extremely positive ($> +3$) AO values on a particular lag

320   day $t$ relative to the SSW central date. Mathematically, these probabilities can be written as $P(AO \mid SSW)$. Per construction, lag day 0 describes the SSW central day. At each lag day, the probabilities are computed by normalizing the number of events fulfilling the respective condition with the total number of available events at the respective lag day (which decreases for large positive and negative lags).

In addition, the overall daily probabilities of AO$< 0$, AO$< -3$ and AO$> +3$ are presented, providing climatological base-

lines $P(AO)$, which are independent of lag time. In any forecast, AO events occur at each day with probabilities of about 49.0% for AO $< 0$, about $0.3\%$ for $AO < -3$ and about $0.1\%$ for $AO > +3$. Asymmetry between positive and negative values arises from the AO distribution that is not perfectly Gaussian (skewness: -0.13).

The fraction of events in the p-SSW composite that have negative AO values fluctuates around $P(AO^{-0} \mid SSW) = 50\%$ at negative lags with only small deviations from the climatology. Within the first week following the event, this fraction

increases and appears to saturate around 60%. Consequently, in the period following a p-SSW, a negative AO is, at each day, approximately 50% more likely compared to a positive AO (60% vs. 40%). The results are consistent between ECMWF and UKMO during the $\pm 4$ week period where the composites for both models consist of more than 30% of all events.

Extremely negative AO values in the dataset appear with a climatological probability that is similar to what would be expected for a (one-sided) 3-sigma-event of a standard normal distribution (0.27%). At negative lags, they occur overall less fre-

quent compared to climatology. In contrast, around lag 0, the probability increases and persists at $P(AO^{-3} \mid SSW) \approx 0.40\%$ for more than four weeks. The increase appears to be larger in the UKMO model, however due to fewer events the diagnostic is also noisier. The fraction of events with extremely positive AO values is smaller compared to climatology throughout the entire lag period. This is largely consistent between the models from ECMWF and UKMO. ERA5 (not shown) overall reveals higher probabilities of negative AO values following SSWs, $P(AO^{-0} \mid SSW)$. However, large uncertainties (95%-CI

$\approx [45\%; 85\%]$) in ERA5 make it difficult to distinguish whether observed differences arise from sampling errors in the reanalysis or from imperfect models. The ERA5 baseline probabilities of AO extremes are modestly lower compared to the S2S models[2] ($P^{ERA5}(AO^{-3}) = 0.06\%$; $P^{ERA5}(AO^{+3}) = 0.02\%$) and not a single AO$^{\pm 3}$ extreme event occurred within a four week period following a real atmosphere SSW, resulting in $P^{ERA5}(AO^{\pm 3} \mid SSW) = 0$, likely due to the very limited sample size.

An altered probability of extreme AO events may be of higher socio-economic relevance than a small shift of the mean. However, the absolute daily probabilities of extremely negative AO events are still small even though the relative increase given the p-SSWs is indeed considerable. In practice, the relevant question might not be how much the probability increases on only one specific day following a p-SSW. It may be more relevant to quantify the increased risk for an extreme AO within a given time period following a p-SSW.

Figure 5 therefore shows the probability of at least one AO$^{-3}$ extreme between day 1 and day $t$ as a function of $t$. We compare the period following p-SSWs, $P(AO^{-3}_{wt} \mid SSW)$ to the respective model climatologies, $P(AO^{-3})$, the ERA5 climatology and

---

[2]Note that we have standardized the AO in ERA5 such that the inter-annual standard deviation is 1, similar to the deseasonalization that is applied to the S2S forecasts. The lower baseline probabilities are consistent with a non-zero kurtosis of the AO distribution in ERA5 of $\sim -0.3$ (ECMWF: $\sim 0.0$, UKMO: $\sim 0.1$).

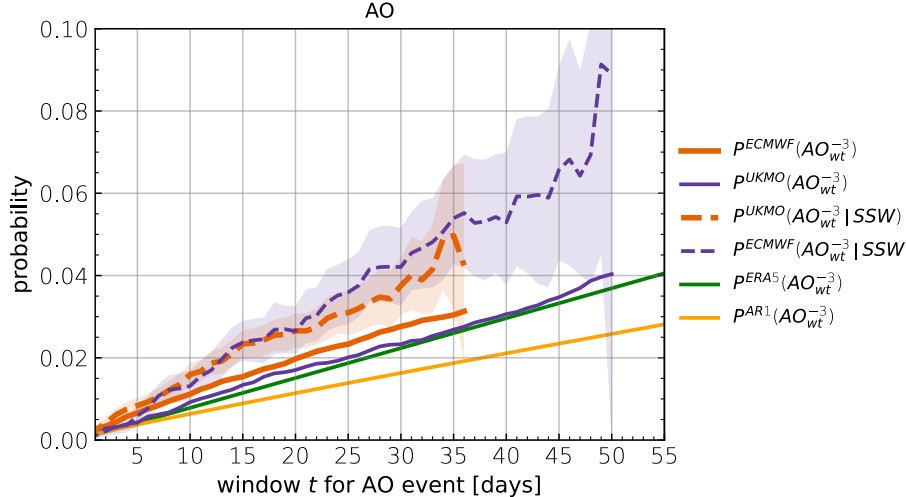

**Figure 5.** Probabilities of at least one $AO^{-3}$ event within a window of time $t$ following p-SSWs (dashed, mean incl. 95%-confidence-interval) are compared to climatology (solid), separately for ECMWF (orange) and UKMO (purple). In addition, the climatologies for ERA5 (green) and a random first-order auto-regressive model of the same 1-day-autocorrelation (yellow) are presented.

an AR1 process of the same autocorrelation as the AO index in ERA5. Confidence intervals were obtained for $P(AO_{wt}^{-3} \mid SSW)$ by bootstrap sampling all SSW events. For ECMWF and UKMO climatology, probabilities were sampled from lead time +10 days[3] to lead time +10+$t$ days within all forecasts. Similarly, baseline probabilities of ERA5 and the AR1-process are obtained by sampling from all days $t_0$ of the time series to day $t_0 + t$, respectively.

Clearly, all probabilities increase with $t$, as the time window for finding at least one $AO^{-3}$ extreme gets wider. However, with increasing $t$, also fewer events contribute to the composite due to the finite forecast lead time, leading to larger sampling errors. The results show that p-SSWs are consistently leading to an increased time-integrated risk of $AO^{-3}$ events. For example, the probability in the ECMWF forecasts of at least one AO extreme within 30 days following the event is 3.8%, compared to 2.9% for its climatology. Overall, p-SSWs seem to affect the probability more in the UKMO model, as the probability following p-SSWs is higher and the climatological baseline is also lower compared to the ECMWF model. The baseline in ERA5 is slightly lower than in the ECMWF model, but agrees well with the UKMO climatology. All probabilities range considerably higher than the probability of a one-sided 3-sigma event for the AR1-process and as before, this is a result of the negative skewness of the AO distribution.

Generally, all probabilities appear approximately linear in $t$, but it should be noted that the linear regime only holds for small enough $t$ as the probability will approach 1 and saturate in the limit of very large $t$. Furthermore, it is expected that for much

---

[3]We choose 10 days as we also start to search for p-SSWs at lead time day 10, however, this choice is arbitrary and the resulting climatology is not very sensitive to this choice.

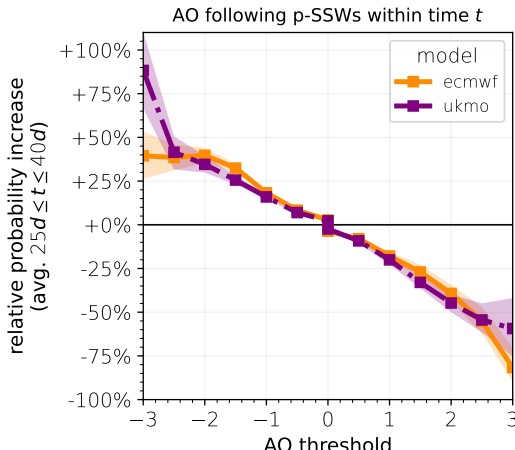

**Figure 6.** Probability increase (in percent) for at least one negative (positive) p-AO extreme below (above) threshold following p-SSWs within a certain period $t$, relative to climatology, averaged over $25\text{d} \leq t \leq 40\text{d}$, separately for ECMWF (orange, solid) and UKMO (purple, dash-dotted).

larger $t$ (which cannot be evaluated here, due to the finite maximum forecast lead time), the effect of a p-SSW increasing the subsequent extreme $\text{AO}^-$ probability diminishes and the climatology will approach the one for p-SSWs.

Based on the presented probabilities, the probability increase of at least one AO event within time $t$ following SSWs can be estimated *relative* to the climatological baseline:

$$\text{relative probability increase} = \frac{P(AO_{wt} \mid SSW)}{P(AO_{wt})} - 1 \tag{1}$$

A relative probability $\frac{P(AO_{wt}|SSW)}{P(AO_{wt})}$ larger than 0 corresponds to an increase of AO probability following SSWs, while negative values describe a probability decrease. This ratio is a function of the length of the time window $t$ (see supplement Fig. S2). In the limit of large $t$, where the SSW influence becomes negligible, it is expected to approach 1, such that the relative probability increase approaches 0. However, for medium time windows $t$ that correspond to a typical timescale of stratosphere-troposphere coupling, the relative probability shows a wide plateau. This motivates the calculation of the relative probability increase averaged over the plateau, which is estimated to correspond to 25 days $\leq t \leq 40$ days, based on Fig. S2. The resulting relative probability increase (Fig. 6) provides an estimate for the extent to which p-SSWs increase the probability of p-AO extreme events – not limited to a specific lag day, but time-integrated and thus independent of $t$. Note that the measure is relative to the climatology, which also includes AO extremes that occur following SSWs. The diagnostic can therefore be interpreted as the relative probability modulation of at least one $\text{AO}^\pm$ event within a certain time period following the occurrence of a SSW, relative to the baseline probability where the stratospheric state is unknown.

The relative probability increase of AO events around 0 (e.g., at least one day below/ above 0) is very small, as these events are already almost certain, even in the climatological reference. Both models show a gradual increase of relative probability of more negative AO thresholds (e.g., $\sim +35\%$ for AO$< -2$) and a gradual decrease for more positive AO thresholds ($\sim -40\%$

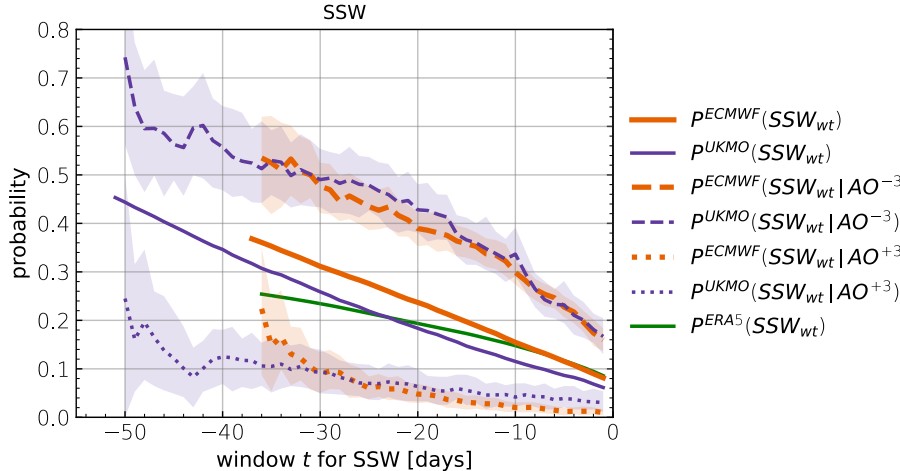

**Figure 7.** Probabilities of at least one day u60 $< 0$ within day $t$ and day -1 relative to day 0, where day 0 is either a randomly sampled day (solid) , an $AO^{-3}$ extreme event (dashed), or an $AO^{+3}$ extreme event (dotted). S2S ECMWF (orange), S2S UKMO (purple) and ERA5 (green).

for AO$> +2$), which is consistent with a shift of the distribution toward more negative values. Quantitative differences in the results between the models are observed for AO thresholds of $\pm 3$. Indeed, sampling uncertainties become considerable for thresholds greater than 2 standard deviations as well, as indicated by 95% confidence intervals that are obtained via bootstrap sampling among all SSW events. However, model discrepancies reach beyond the indicated confidence intervals, which will be briefly discussed in section 8.

## 6 Toward attribution of predicted AO extremes to preceding SSWs

The last section focused on given p-SSWs and subsequent statistical signatures in AO extremes within a period $t$: $P(AO_{wt} \mid SSW)$. It was shown that AO$^-$ extremes are significantly more likely following a SSW.

In this section, we aim to evaluate the alternative question: How many AO$^-$ events may statistically be attributable to preceding SSWs?

AO$^-$ extremes occur with and without preceding SSWs. As outlined in subsection 3.5, the distinction of whether an AO extreme was or was not exposed to a preceding stratospheric extreme requires choosing a time window for the potential exposure (e.g., was a given AO extreme preceded by a SSW within the preceding 30 days or not).

The basis of the evaluation in this section is that instead of conditioning on the occurrence of a SSW, we condition on the occurrence of an AO extreme. This allows the classification of all AO events according to whether they were or were not exposed to a preceding SSW within a time window $t$. In total, the ECMWF analysis is based on 752 $AO^{-3}$ and 486 $AO^{+3}$ events, where asymmetry arises from non-zero skewness of the AO distribution (UKMO: 299 and 186).

Fig. 7 shows the probability that $AO^{\pm 3}$ events are preceded by at least one day of negative u60 within time $t$, corresponding to $P(SSW_{wt} \mid AO^{\pm 3})$. For example, the probability of p-SSW occurrence within 30 days preceding $AO^{-3}$ extremes is close to 0.5 in both models, whereas it is around 0.1 preceding $AO^{+3}$ extremes. 95% confidence intervals, which were derived by bootstrap resampling all AO events, confirm that the diagnostics get less robust for larger time windows, due to fewer available events contributing to the AO composite. The probabilities of the extremes to be *not* preceded by at least one day of negative u60 are given by $P(\neg SSW_{wt} \mid AO^{\pm 3}) = 1 - P(SSW_{wt} \mid AO^{\pm 3})$.

We can use the estimated probabilities $P(SSW_{wt} \mid AO^{\pm 3})$ to evaluate the fraction of attributable risk (FAR) of $AO^-$ events to preceding SSWs as follows. Note that in this study we neglect potential common drivers of both AO and stratospheric extremes, such as due to tropical teleconnections. Consequently our analyses of FAR may overestimate the part that is solely due to the stratosphere. Nevertheless, they serve to quantify the statistical association between stratospheric extremes and the AO, as well as quantify the predictive skill due to the stratosphere.

First we define the FAR among the exposed[4]:

$$\text{FAR}_e = \frac{\text{risk among the exposed} - \text{risk among the unexposed}}{\text{risk among the exposed}} \tag{2}$$

This quantifies the fraction of SSW–$AO^-$ co-occurrences ("exposed" category) in addition to fortuitously aligned events, where the latter risk in the numerator is given by $P(AO^- \mid \neg SSW_{wt})$. An $\text{FAR}_e$ of 0 means that the probability of finding an $AO^-$ extreme is independent of exposure to a preceding SSW. Likewise, an $\text{FAR}_e$ of 1 means that $AO^-$ extremes do not happen without exposure to a preceding SSW. We can estimate the involved probabilities of $AO^-$ events exposed or not to a preceding SSW using Bayes theorem:

$$P(AO^- \mid SSW_{wt}) = \frac{P(SSW_{wt} \mid AO^-) \cdot P(AO^-)}{P(SSW_{wt})} \tag{3}$$

$$P(AO^- \mid \neg SSW_{wt}) = \frac{P(\neg SSW_{wt} \mid AO^-) \cdot P(AO^-)}{P(\neg SSW_{wt})} = \frac{[1 - P(SSW_{wt} \mid AO^-)] \cdot P(AO^-)}{1 - P(SSW_{wt})} \tag{4}$$

Inserting these expressions we obtain for $\text{FAR}_e$:

$$\text{FAR}_e = \frac{P(AO^- \mid SSW_{wt}) - P(AO^- \mid \neg SSW_{wt})}{P(AO^- \mid SSW_{wt})} = 1 - \frac{P(SSW_{wt})}{P(\neg SSW_{wt})} \frac{P(\neg SSW_{wt} \mid AO^-)}{P(SSW_{wt} \mid AO^-)} \tag{5}$$

This expression involves $P(SSW_{wt})$, which represents the baseline climatology of the probability that any random day (i.e., regardless of its AO value) is preceded by a SSW within time $t$ (full lines in Fig. 7). By definition, $P(\neg SSW_{wt}) = 1 - P(SSW_{wt})$.

Our estimates of $\text{FAR}_e$ are shown in Fig. 8a as a function of time window $t$, for two AO event thresholds (–2 and –3). We find that these estimates are not a strong function of the chosen time window. Fig. 8b summarizes the $\text{FAR}_e$ averaged over time windows of 25 to 40 days: For example, based on the ECMWF forecasts we estimate that on average about 50% of all $AO^{-3}$ events that are preceded by a SSW may statistically be attributable to that SSW. For the UKMO forecasts this value is slightly

---

[4]$\text{FAR}_e$ is commonly used in climate attribution science, e.g., to determine the likelihood that an extreme weather event is attributable to anthropogenic climate change (see, e.g., Allen, 2003; Stone and Allen, 2005; Stott et al., 2016).

higher (∼60%). For $AO^{-2}$ events these percentages are somewhat smaller but overall similar between the models. Boxplots reveal that associated sampling uncertainties are generally small, but larger for $AO^{-3}$ events.

The attributable risk may also be evaluated for *any* $AO^-$ extreme (from the entire population). In this case one is interested in quantifying the fraction of $AO^-$ extremes that occur in addition to those that are "unexposed" (were not preceded by a SSW). The corresponding FAR among the population is defined as:

$$\text{FAR}_p = \frac{\text{risk among the population} - \text{risk among the unexposed}}{\text{risk among the population}} = \frac{P(AO^-) - P(AO^- \mid \neg SSW_{wt})}{P(AO^-)} = 1 - \frac{P(\neg SSW \mid AO)}{P(\neg SSW)}$$

(6)

where the corresponding expressions from Bayes theorem have been inserted as before. $\text{FAR}_p$ then also quantifies the fraction of AO extremes that may statistically be attributable to a preceding SSW. For example, an $\text{FAR}_p$ of 0 means that SSWs do not increase the probability of AO extremes, whereas an $\text{FAR}_p$ of 1 means that all AO extremes may be attributable to a preceding SSW within time $t$. The same caveats about common drivers as for $\text{FAR}_e$ should be kept in mind.

Figure 8c shows our estimates of $\text{FAR}_p$ as a function of time window $t$, similar as for $\text{FAR}_e$. As expected, estimates of $\text{FAR}_p$ are generally lower than for $\text{FAR}_e$: the likelihood of any AO extreme to be attributable to a SSW that may or may not have happened before the AO extreme should be much smaller than that of an AO extreme that was indeed preceded by a SSW. $\text{FAR}_p$ increases somewhat with $t$ for small $t$, but tends to saturate for windows longer than about 2 weeks. For $AO^{-2}$ events both models saturate near 0.2, whereas for $AO^{-3}$ events they show slightly larger $\text{FAR}_p$ of around 0.25-0.3. Overall our estimates therefore suggest that between 20-30% of $AO^-$ extremes may statistically be attributable to a preceding SSW (within 2-6 weeks). Fig. 8d summarizes the $\text{FAR}_p$ averaged over time windows of 25 to 40 days. Despite the lower number of contributing events for larger time windows, associated sampling uncertainties are small (e.g., 95% confidence intervals for $\text{FAR}_p$ in ECMWF for $AO^{-3}$: $[21\%; 28\%]$).

## 7 Strong polar vortex events and associated AO extremes

The previous sections revealed that SSWs increase the probability of subsequent $AO^-$ extremes and that a significant fraction of $AO^-$ extremes may be attributable to preceding SSWs. In the following, we summarize an analogous analysis for the statistical relationship between strong polar vortex events (SPVs) and $AO^+$ extremes.

The composite-mean evolution of p-SPVs (Fig. 9) reveals that u60 anomalies are of opposite sign, somewhat weaker in magnitude, but otherwise qualitatively similar to p-SSWs (lag 0: $\sim +20$ ms$^{-1}$ for p-SPVs; $\sim -30$ ms$^{-1}$ for p-SSWs, cf. Fig. 2). Both S2S models agree very well in this respect. Moreover, for negative lags, there is little difference compared to a corresponding composite based on ERA5 data, but for positive lags, u60 is slightly stronger in ERA5. The NAM response at 200hPa and 1000hPa (=AO) is qualitatively similar for p-SPVs and p-SSWs (with opposite sign), but the anomalies are again slightly weaker for p-SPVs, which is consistent with the weaker u60 anomalies (lag 21: +0.35 at 200hPa, +0.25 at 1000hPa). It is interesting that the NAM200 seems to react later to p-SPVs than to p-SSWs: While the index for p-SSWs starts to shift significantly to negative values already at lag $-10$ on average, a shift to positive NAM200 values for p-SPVs is observed

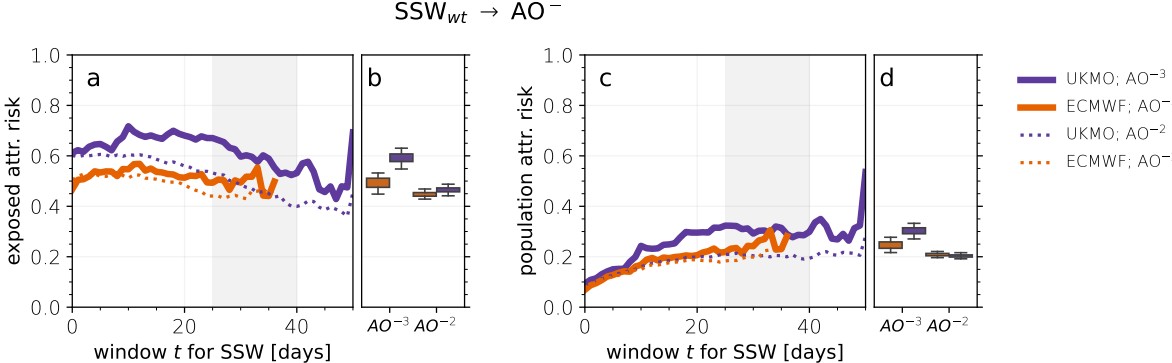

**Figure 8.** Left: Fraction of $AO^{-2}$ (dotted) and $AO^{-3}$ (solid) extremes that are preceded by a SSW within time $t$ that may be attributable to the SSW (fraction of attributable risk among the exposed/ $FAR_e$, panel a). Boxplots (quartiles 1 to 3 and 95% confidence intervals, obtained via bootstrap resampling) show $FAR_e$ averaged over time windows 25 to 40 days (gray shaded), as function of AO threshold (panel b). Right: Fraction of all $AO^{-2}$ and $AO^{-3}$ extremes that may be attributable to a preceding SSW within time $t$ (fraction of attributable risk among the population/ $FAR_p$, panel c). Boxplots (as in panel b) show $FAR_p$ averaged over time windows 25 to 40 days (panel d). Note that for larger $t$, fewer events contribute to the diagnostics, hence, observed fluctuations for long time windows $t$ are likely related to sampling uncertainty. UKMO (purple) and ECMWF (orange).

only from lag $-5$ on. As with p-SSWs, the evolution of the NAM at 200hPa and 1000hPa relative to p-SPVs is less robust in ERA5 due to the smaller sample size, however, the anomalies tend to be slightly more pronounced than in the two S2S
models. Overall, the composite-mean evolution of p-SPVs in the ECMWF and UKMO models appear to be consistent with real-atmosphere SPVs (as revealed by reanalysis data), as well as with previous studies (e.g., Baldwin and Dunkerton, 2001).

Following the same methodology as for p-SSWs, we use the large event sample sizes to quantify the statistical relation between p-SPVs and subsequent $AO^+$ extremes. First, we quantify the relative probability increase for at least one AO extreme after a given p-SPV within a certain time. Second, we analyze how many $AO^+$ extremes may be attributable to preceding
p-SPVs.

Figure 10 shows the relative probability increase of AO extremes following SPVs relative to climatology as a function of the AO threshold, for both S2S models and averaged over time windows 25 days $\leq t \leq 40$ days:

$$\text{relative probability increase} = \frac{P(AO_{wt} \mid SPV)}{P(AO_{wt})} - 1 \tag{7}$$

Consistent with the positive shift of the AO distribution following SPVs, the risk gradually increases for positive AO ex-
tremes, whereas it gradually decreases for negative AO extremes. For extreme thresholds of up to 2 standard deviations, the relative probability change appears to be of similar magnitude compared to periods following SSWs ($\approx$30-40%, see Fig. 6). Larger thresholds reveal a reduced probability change compared to SSWs, however, 95% confidence intervals mark increasing sampling uncertainty, especially for $AO_{wt}^{+3}$ events.

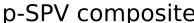

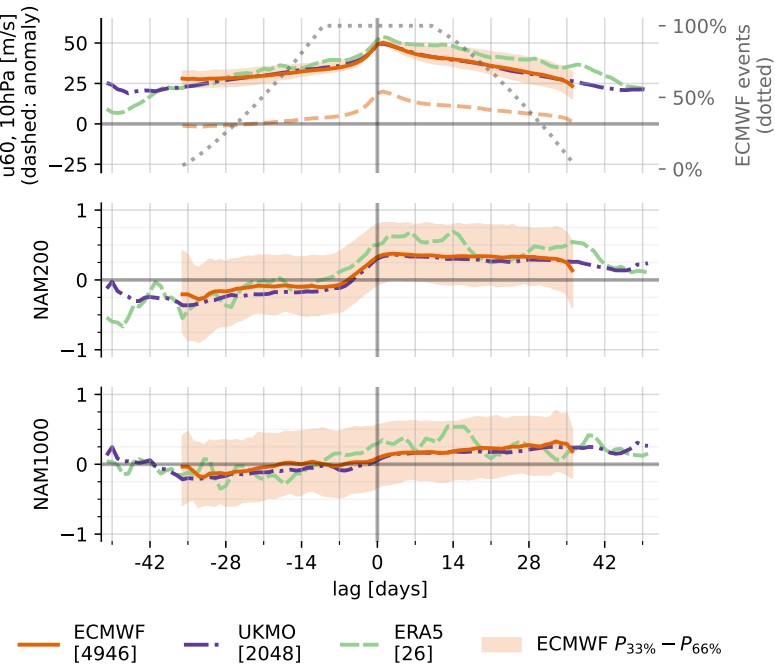

**Figure 9.** As in Fig. 2, for p-SPVs.

Figure 11 shows our estimates of the fraction of positive AO extremes that may be attributable to a preceding p-SPV within
a time period $t$:

$$\text{FAR}_e = \frac{P(AO^+ \mid SPV_{wt}) - P(AO^+ \mid \neg SPV_{wt})}{P(AO^+ \mid SPV_{wt})} \tag{8}$$

$$\text{FAR}_p = \frac{P(AO^+) - P(AO^+ \mid \neg SPV_{wt})}{P(AO^+)} \tag{9}$$

where $\text{FAR}_e$ and $\text{FAR}_p$ denote exposed and population attributable risk, as in section 6 for SSWs and $AO^-$ events. Among
all $AO^{+3}$ events that are preceded by at least one SPV event within four weeks, about 55% (UKMO) to 65% (ECMWF) may
be attributable to the SPV (Figs. 11a, 11b). However, significant sensitivities to the exact time window are observed, as well
as differences between the models. One problem is the strong seasonal dependence of SPV events, as most events occur in
December when the polar vortex is generally strongest. AO extremes that happen later in the winter have therefore a smaller
probability to be preceded by a SPV event within a short time window than AO extremes that occur in December or January.
$AO^{+2}$ events reveal a fraction of attributable risk among the exposed to preceding SPVs of around 40% to 55%, similar to
SSWs and $AO^{-2}$ events.

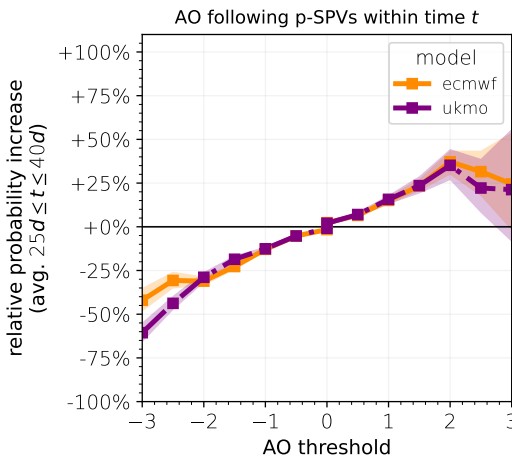

**Figure 10.** As in Fig. 6, for p-SPVs and subsequent AO extremes within time $t$.

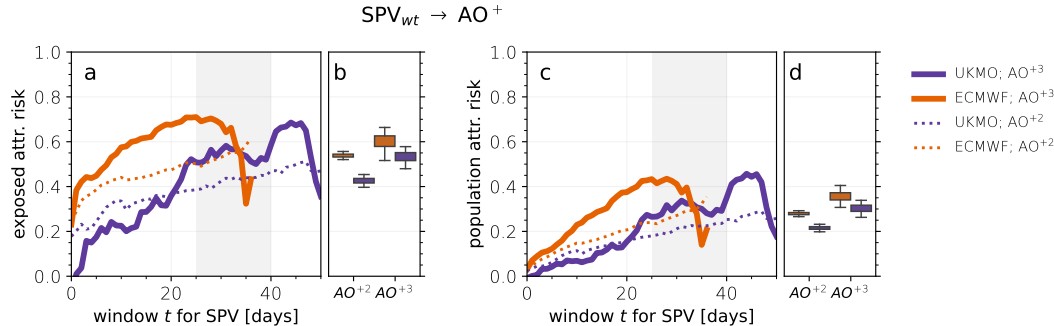

**Figure 11.** As in Fig. 8, for positive AO extremes that may be attributable to preceding SPV events within time $t$.

Finally, the fraction of all $AO^+$ extremes that may be attributable to preceding SPVs is slightly larger but similar to that for $AO^-$ extremes and SSWs, with a population attributable risk of around one quarter for $AO^{+2}$ and around one third for $AO^{+3}$ extremes for preceding time windows of 25 to 40 days (Figs. 11c, 11d).

More detailed analyses that apply the diagnostics presented in Fig. 3, Fig. 4 and Fig. 5 to positive AO extremes and p-SPVs

are shown in the supplement.

## 8  Conclusions

Our results, based on a large number of extended-range ensemble forecasts, provide further evidence for stratospheric modulation of large-scale weather patterns near the surface, broadly consistent with previous results (Domeisen and Butler, 2020b, and references therein). Previous studies generally suffer from relatively small available sample sizes, which hampers estimation

of robust statistical relationships between stratospheric and tropospheric extremes (= rare events). In this study, by analyzing extended-range forecast periods around predicted extreme events (e.g., p-SSWs), we effectively boost the available sample size by more than a factor of 100 and are therefore in the position to obtain robust estimates in response to our research questions:

1. By how much is the probability of persistently positive or negative AO phases increased following stratospheric polar vortex extremes?

Climatologically, 38% of negative AO phases (days with consecutive AO$< 0$) are longer than 7 days. Following p-SSWs, this is increased to 44%, which corresponds to a relative increase of 16%.

Following p-SPVs, the probability of positive AO phases that last longer than 7 days is increased from 40% to 44%.

2. By how much is the probability of subsequent AO extremes increased following stratospheric polar vortex extremes?

Following p-SSWs, the probability of subsequent negative AO extremes increases whereas it decreases for positive AO
extremes. For instance, AO$^{-3}$ events are about 40% (ECMWF forecasts) to about 80% (UKMO forecasts) more likely following p-SSWs. However, the absolute probabilities are still low, i.e., only 3.5% of SSWs are followed by AO$^{-3}$ within four weeks, based on ECMWF forecasts (UKMO: 4%).

Following p-SPVs, the probability of AO$^{+3}$ is increased by about 25% relative to climatology, whereas AO$^{-3}$ occur about 40% (ECMWF) to 60% (UKMO) less often.

3. What fraction of AO extremes may be attributable to preceding stratospheric polar vortex extremes?

About 50% (ECMWF) to 60% (UKMO) of AO$^{-3}$ extremes that occur following a SSW may be attributable to that SSW (fraction of attributable risk among the exposed). 20-30% of all AO$^{-3}$ events may be attributable to preceding SSWs (fraction of attributable risk among the population). "Attributable" does not necessarily imply strict causality (see discussion below), but here refers to the fraction of SSW-AO$^-$ co-occurrences in addition to fortuitously aligned events.

While our stratospheric event definitions are based on absolute thresholds of the zonal-mean zonal wind, the tropospheric response is quantified via standardized anomalies of averaged geopotential. The construction of an appropriate corresponding climatology is crucial, in particular for the analysis of extreme events. However, it is also not unambiguous. Standardized anomalies are computed by normalizing differences from a population mean with the population standard deviation (taking into account seasonal variations). As the population is usually finite, any additional data point may change the population mean
and will change the population standard deviation, resulting in a small adjustment of all previous (standardized) data points. On the one hand, the effect is negligible in the limit of a large population. On the other hand, it is generally larger when the additional data point is an outlier with respect to the previous distribution. For this study, S2S forecasts were deseasonalized using the available hindcasts. The assumption is that these hindcasts sufficiently sample different kinds of variability, such that a) extreme events that occurred in individual years do not significantly distort the population distribution and thereby also the
population mean and standard deviation and that b) the constructed population is robust across different initialization dates (e.g., a given event that is equally predicted at two different leadtimes corresponds to a the same standardized event in both

model integrations).

Do the analyses of modulated probabilities allow conclusions about causal links between stratospheric and tropospheric circulation extremes?

A definition of (probabilistic) causality is provided by Pearl (2009):

$$P(\text{effect} \mid do(\text{cause})) > P(\text{effect} \mid do(\neg\text{cause})), \tag{10}$$

where the $do$ operator denotes an intervention that forces the occurrence or not-occurrence of the cause[5]. In the atmosphere, such controlled situations can usually only be simulated using numerical model experiments. In this study, a post-hoc analysis of an existing dataset is presented. No interventions are performed and therefore, no strict causal relations can be inferred following the provided definition. Instead, conditional probabilities are computed, which Pearl (2009) calls a predictive or observational approach, e.g.:

$$P(AO^- \mid SSW) > P(AO^- \mid \neg SSW). \tag{11}$$

Our knowledge of coupled stratosphere-troposphere dynamics suggests that a causal connection does in principle exist[6]. This connection manifests in observed conditional probabilities, which may, however, be modulated also by further possibly involved pathways.

First, conditional probabilities may in practice overestimate the (direct) causal link between stratospheric and AO extreme due to the existence of confounding factors (see scenario $c$ listed in the introduction). For example, the Madden-Julian Oscillation (MJO) may lead to modified risk of AO extremes (Barnes et al., 2019) while at the same time modifying the likelihood of SSWs (Garfinkel et al., 2012). On the other hand, the dynamical coupling between the MJO and the AO may involve a stratospheric pathway (Garfinkel et al., 2014) and in such cases the stratosphere does represent a causal driver of AO modulations. Similar arguments hold for impacts due to climate variability, such as Arctic sea ice concentrations (Kretschmer et al., 2016) and the El Nino Southern Oscillation (ENSO) (Domeisen et al., 2019). Causal pathways may in such cases be disentangled using a causal inference-based network (Kretschmer et al., 2021). We have carried out preliminary analyses using such a framework to distinguish causal pathways during different ENSO phases, which suggest that the direct pathway *polar vortex* $\rightarrow$ *AO extremes* is significantly stronger than those via ENSO. A detailed analysis of these pathways is left for future work.

However, even if common drivers can be neglected the statistical nature of inferred fraction of attributable risk can only quantify an *effective* causality in the following sense. Assume, for the moment, that all SSWs cause an $AO^-$ extreme, but $AO^-$ extremes additionally occur due to internal tropospheric variability. In this case some of the observed $AO^-$ extremes may have happened due to internal tropospheric variability alone while additionally be forced/enhanced by a preceding SSW (see scenario $b$ listed in the introduction). A probability analysis (e.g., estimating the FAR among the population) will then always

---

[5]This definition relies on counterfactual dependence, i.e., if there had not been the cause, then there would not have been the effect (and if there had been the cause, then there would have been the effect).

[6]It is important to keep in mind that the coupling is, in general, mutual and causality works in both directions (even though, as always, any cause has to precede the effect).

underestimate the actual causal link and can only reveal an effective causality. This also represents a limitation of the binary classification (AO extreme / no AO extreme).

Despite these caveats, conditional probabilities may provide useful insights. The conversion into statistical metrics such as
RPI and FAR may thereby facilitate the practically relevant interpretation. For example, RPI of AO extremes due to the prior occurrence of a stratospheric extreme does serve to quantify the state of the stratosphere as a predictor of subsequent AO extremes, which may be of practical value regardless of its underlying causal nature. Furthermore, FAR provides an estimate of how many AO extremes would statistically be expected less without preceding stratospheric events, when keeping in mind that "without a preceding stratospheric event" would require to remove also confounding factors.


How should the observed differences between ECMWF and UKMO model be interpreted? Overall, our analyses show that the probability modulation of AO extremes up to about two standard deviations given preceding stratospheric extremes are similar between the ECMWF and the UKMO model. AO extremes of three standard deviations, i.e., $AO < -3$ and $AO > +3$ reveal discrepancies between the models. Our bootstrapping approach, e.g., for the relative probability increase (Fig. 6), shows
that especially analyses based on UKMO forecasts become less robust. However, the observed discrepancies cannot be solely attributed to sampling uncertainty, given that they exist also beyond the respective 95% confidence intervals. Which model better represents the dynamics of the real atmosphere is difficult to assess, as the observational record is too short to allow for robust, similar analyses. Potential causes of the observed differences are numerous, involving differences in wave-mean flow feedbacks or external forcings, e.g., from the tropics. Augier and Lindborg (2013) show that the eddy kinetic energy
spectrum in the ECMWF model is still in parts unrealistic and that the model may be too dissipative even at large scales, clearly indicating that models are unable to reproduce real-atmosphere dynamics perfectly accurate. Lawrence et al. (2022) investigate biases in different S2S models and find, inter alia, a modest cold bias in the ECMWF and a modest warm bias in the UKMO model in the extra-tropical lower stratosphere. As the lower stratosphere has been shown to play an important role in stratosphere-troposphere coupling, we speculate that occurrences of tropospheric extremes following stratospheric circulation
anomalies are sensitive to temperature biases in this region. However, a detailed analysis would be beyond the scope of this study.

In general, we note that any two different imperfect models, will likely always reveal quantitative differences in the analysis of extreme events for a sufficiently strict extreme threshold. In the present study, we find such differences, e.g., for the relative risk, at a threshold of around three standard deviations. It is possible that more data are needed to conclusively attribute the
differences to particular dynamical processes. Nevertheless, we argue that our analyses, even at a threshold of 3 standard deviations and given the associated uncertainties, are able to provide insightful quantitative estimates; especially as no obvious a priori estimate exists even for the order of magnitude of the investigated probability metrics.

In addition to the particular points already mentioned, future work should address the question, how much of the predicted
surface impact following predicted stratospheric extremes, i.e., following p-SSWs and p-SPVs, can be explained by the AO.

Lastly, we conclude that the analysis of *predicted* events offers potential for improved statistical characterization of other atmospheric extreme events, provided that the forecast model is capable of truthfully representing the event of interest.

*Data availability.* Forecasts from the S2S archive can be found at https://apps.ecmwf.int/datasets/data/s2s. ERA5 data is available at https://cds.climate.copernicus.eu/cdsapp#!/dataset/reanalysis-era5-pressure-levels.

## Appendix A: Deseasonalization of S2S Forecasts

In addition to realtime forecasts, all S2S forecasting systems create also hindcasts (or "reforecasts"), which allow the construction of the respective model's climatology. In the following, we describe the procedure[7] we applied to compute a climatology of a forecast that starts on some date $d$ (month & day of month).

1. Compute the ensemble mean of the hindcasts (Fig. A1a).

2. Compute the inter-annual mean of the hindcast ensemble means. In case of the ECMWF forecasts for example, the hindcasts cover the past 20 years (see Fig. A1b).

3. Select all (inter-annually averaged) hindcasts that start within $\pm 14$ days relative to the date $d$ (the start of the forecast of interest). In case of the ECMWF model, this selection subsumes 9 (inter-annually averaged) hindcasts, since hindcasts are available for every Monday and Thursday (see Fig. A1c).

4. Average the hindcasts obtained in 3, such that the forecast valid times match (e.g., average forecasts for Feb 22, Feb 23, ... as opposed to matching forecast lead times, e.g., forecasts with lead time +4, +5, ..., see Fig. A1c).

5. Apply, to the resulting time-series, a 7-day running mean filter (Fig. A1d).

6. Due to the $\pm 14$ day window, the resulting time-series starts earlier than date $d$ and covers a period that is longer than the forecast of interest. Cut the time-series at the beginning and at the end such that it matches the time-series of the forecast of interest. This gives the climatology (see Fig. A1d).

Anomalies are obtained by subtracting the climatology from the raw field. Standardized anomalies can be computed by dividing the anomalies through a climatology standard deviation, which is computed similar to the climatological mean, but where

– *(ad step 1)* instead of the ensemble mean, the unperturbed control run is selected (or any other single ensemble member). Using the ensemble mean would result in a too small inter-annual standard deviation at long forecast lead times (see step 2), because at long lead times, the ensemble mean *always* tends to the climatological mean state.

---

[7]based on the ECMWF article "Re-forecast for medium and extended forecast range" (https://www.ecmwf.int/en/forecasts/documentation-and-support/extended-range/re-forecast-medium-and-extended-forecast-range, accessed on 23 Aug 2021).

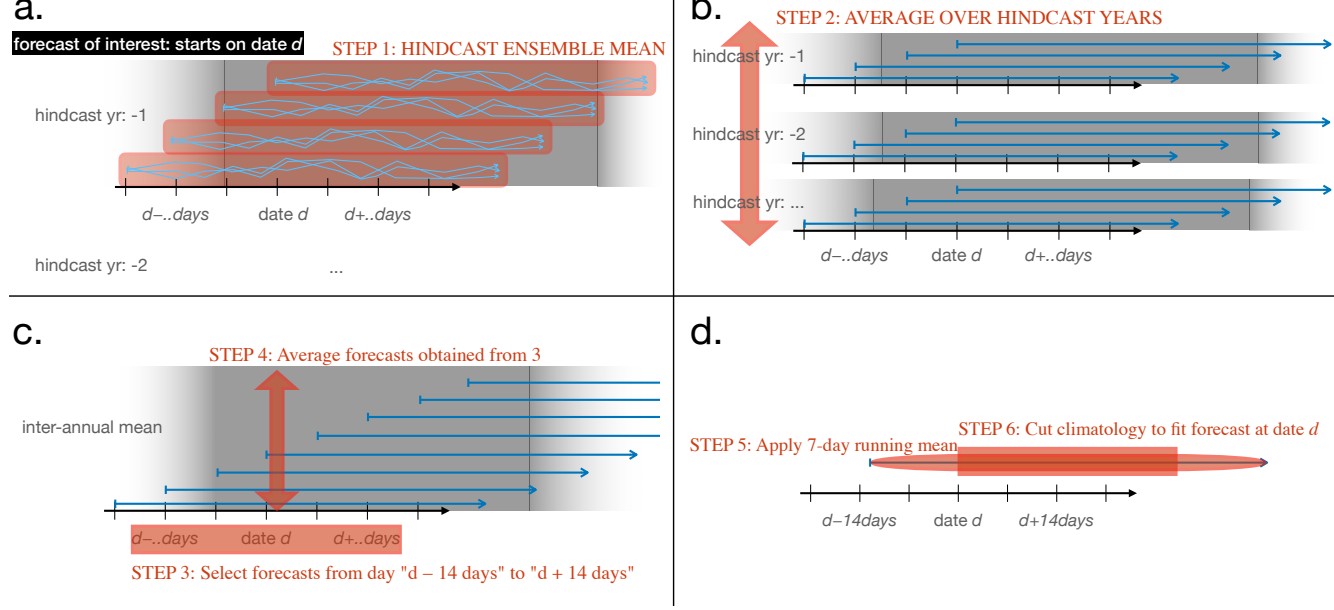

**Figure A1.** Schematic workflow for the computation of a climatology for a S2S forecast model, based on hindcasts. Gray planes illustrate that forecasts belong to the same hindcast year, where the axis from left to right denotes time and the axis from the front to the back.

- *(ad step 2)* instead of the inter-annual mean, the inter-annual standard deviation is computed.

The presented deseasonalization procedure comes with several implications, for example:

- The climatologies for realtime forecasts and for hindcasts are always based only on hindcasts.

- By computing anomalies from a climatology, model errors that are a function of the season, are mitigated.

- By computing anomalies from a climatology, model errors that are a function of the forecast lead time ("model drift"), are not mitigated, because the climatology averages information that stems from different forecast lead times (see step 4).

- In case of the ECMWF model, 9 hindcast ensembles / four-week-window · 20 years · 11 ensemble member = 1980
integrations contribute to the construction of one climatology.

## Appendix B: P(SSW) proxy

From observations, the annual probability of SSWs can be derived by normalizing the number of winters with SSWs with the total number of winters. In the S2S model framework, it is however less straightforward to compute the frequency of SSWs per winter, as the maximum leadtime is shorter than a winter period and many forecasts overlap. It is reasonable to tie a 0%

SSW-probability to the case where there is not one ensemble member in any of the forecasts that predicts a SSW. The 100% upper boundary is less clear: Should the probability be 100% if all ensemble members in all forecasts show a SSW? In that case, a longer maximum leadtime would result in a higher SSW-probability even for the same model. Should the probability be 100% if there is at least one ensemble forecast in a winter where all members show a SSW? Again, the result would depend on the ensemble size, i.e., the technical setup, not solely on the model physics.

In this study, we compute a proxy for the model's seasonal SSW probability based on the number of SSWs per forecast day, as described in the following:

For each winter season $i$, forecasts with initialization dates between mid-November and mid-February are analyzed, resulting in a total of $\tilde{N} = \sum_i \tilde{N}_i$ forecast runs (counting ensemble members separately). We search for p-SSWs only in forecasts that have solely positive u60 within the first 10 days after initialization, resulting in $N = \sum_i N_i$ forecasts ($N \leq \tilde{N}$). We find $E_i$ p-

SSW events in the winter seasons, respectively, and group those by daily leadtime (similar to Fig. 1, bottom left panel), yielding $E_{i,d}$ p-SSWs in winter $i$ at leadtime $+d$ days. As $E_{i,d}$ is approximately constant over leadtime, we compute the average number of p-SSWs in winter $i$ per day leadtime: $E_i = \overline{E_{i,d}}$, where the overbar denotes the mean over lead times. Hence, the probability that a random forecast in winter $i$ at a random leadtime shows a p-SSW is $p_{i,daily} = \frac{E_i}{N_i}$. The probability of no SSW for an entire winter ($\approx 135$ days from mid-November to end of March) is therefore $(1 - p_{i,daily})^{135}$. Finally, the probability of at least

one SSW in winter $i$ becomes: $p_i = 1 - (1 - p_{i,daily})^{135}$, as presented in Fig. 1 (top left panel). The model's average seasonal SSW probability becomes $p = [p_i]$, where the brackets denote the average over different seasons.

Note that the computed probabilities $p$ and $p_i$ quantify the model's tendency to predict SSWs. Particularly, this allows for inter-annual comparison and comparison between different models. However, the probabilities themselves require careful interpretation, which is why we refer to a SSW probability "proxy". Note that

– the probability quantifies SSW occurrences beyond 10 days leadtime. Thus, inter-annual variations of SSW probabilities arise only from phenomena that are predictable at more than 10 days ahead. This is also the main reason why real atmosphere SSWs have only limited effect on the computed SSW probability.

– the SSW probability becomes 0% if there are no ensemble members that predict SSWs at any time beyond 10 days leadtime. A 100% probability is only reached if all ensemble members predict SSWs at each day leadtime. Fig. B1

shows the analytical relation between daily probability $p_{i,daily}$ and the associated seasonal probability $p_i$. For instance, a daily probability of 2% already leads to a seasonal probability of about 90%. In addition to the analytical relation, the probabilities are shown for all seasons as derived from the ECMWF forecasts.

– seasonality is not explicitly resolved in the calculations, but assumed to average out when enough forecasts are sampled.

*Author contributions.* JS performed the analyses under the guidance of TB. JS wrote the first draft of the manuscript. Both authors con-
tributed to the interpretation of the results and improved the manuscript.

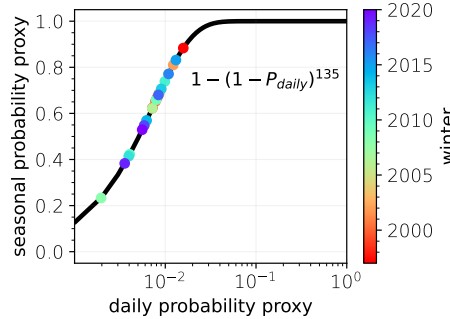

**Figure B1.** Estimating a seasonal SSW probability proxy based on daily SSW probabilities. Colored points show the computed seasonal probability proxy for different winter seasons as applied to the ECMWF forecasts.

*Competing interests.* The authors declare that they have no conflict of interest.

*Acknowledgements.* The authors thank Inna Polichtchouk for fruitful discussion on deseasonalization of S2S data. TB has been supported by the German Research Foundation (DFG) (grant no. SFB/TRR 165; Waves to Weather project). JS further appreciates the valuable scientific exchange within Waves to Weather's early career scientist program. This work is based on S2S data. S2S is a joint initiative of the World Weather Research Programme (WWRP) and the World Climate Research Programme (WCRP). The original S2S database is hosted at ECMWF as an extension of the TIGGE database. Finally, we thank Sandro Lubis and the second, anonymous reviewer for their constructive comments that helped to improve the manuscript.

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
