# Peer review of "Stratospheric Modulation of Arctic Oscillation Extremes as Represented by Extended-Range Ensemble Forecasts"

_Weather and Climate Dynamics, 2021_

## Author Comment (AC1)

We would like to thank both referees for carefully reading our manuscript and we appreciate the constructive feedback. In addition, we would like to thank the editor, Nili Harnik.

Two important issues that both referees highlighted to us are the 1) differences in the results between the two models, and 2) the methodology, particularly with regard to interpretation in terms of causality. Therefore, our response below begins with a general discussion regarding these two issues. Thereafter, we address the specific concerns and suggestions raised by the reviewers and outline how we plan to adapt the manuscript, accordingly.

The referees' comments are in *italic gray,* our responses are in blue.
* * *
Model Differences

We agree that the differences between the two models are interesting and that they need to be considered for the interpretation of the results. However, we note the following points that may help to put the observed differences into perspective.

Generally, the analysis of extreme events is expected to be highly sensitive to (tiny) modulations of the underlying distribution. Our results show that both models agree well in terms of the *average* AO shift following stratospheric events (Figs. 2, 9). Differences by a factor of about 2 are observed for the analysis of extreme events (e.g., Fig. 6). However, we believe that these discrepancies are not overly severe, since small changes in the distributions' tails could translate quickly into different orders of magnitude.

Furthermore, the differences between the models come out most prominent in Figs. 6 and 10, where the relative risk increase of AO–/ AO+ is presented following SSWs/ SPVs. However, we note that particularly the relative risk increase is very sensitive to changes in the climatological occurrence of AO extremes, which appears as a small number in the denominator. With this in mind, we plan to include the *absolute risk difference*, which may complement the analysis.

In discussing potential physical processes that could explain the observed model differences, we will consider, e.g., the reference noted to us by the editor. However, a detailed analysis of the model differences is beyond the scope of our study.

Methodology and causality

We appreciate the reviewers' valuable feedback, which has helped us to clarify a few important points. We agree that the manuscript will benefit from an improved description and a more accurate interpretation of the methodology. Therefore, we plan to take the following steps:
I.   We will provide an overview table that summarizes our event definitions and shows how conditional probabilities are derived from the forecasts.
II.  Based on the conditional probabilities that are stated in the table, we will provide equations for the diagnostics displayed in Figs. 5, 6, 7, 8, 10, 11. Where appropriate, we will adapt the terminology to be consistent with language that is used for similar problems in other fields (e.g., "relative risk increase" and "(absolute) risk difference" in medical statistics).
III. We recognize that the interpretation of our questions 2 and 3 need to be carefully separated in terms of causality.
     *ad Q2*: We found that stratospheric events enhance the probability for AO extremes relative to climatology, therefore, the stratospheric event can be considered a (statistical) predictor. As noted by the referees, our analysis is not proof of causality in a strict sense because other processes could be involved. We are currently in the process of analyzing the role of such processes, for instance, SST variability associated with ENSO and plan to include relevant conclusions from this analysis in the discussion. However, a detailed analysis of further processes is beyond the scope of the present study and is left for future work.
     *ad Q3*: We have analyzed how often AO extremes are preceded by certain stratospheric events and compared the number of observed events to the number of expected events based on climatology. We recognize that this analysis is not able to prove causality, even in the absence of other processes. While we think that a quantification of the exceedance

occurrence of stratospheric events ahead of AO extremes is still useful, it requires a fundamentally different interpretation compared to our analysis of Q2. Instead of "caused by p-SSW"/ "caused by u60<0" (e.g., Figs. 8, 11) we will refer to the exceedance probability of preceding p-SSWs/ u60<0.

**Response to comment by anonymous referee #1**

*The article by Jonas Spaeth and Thomas Birner entitled "Stratospheric Modulation of Arctic Oscillation Extremes as Represented by Extended-Range Ensemble Forecasts" discusses the influence of the stratospheric polar vortex on the surface climate. This topic has been discussed a lot in the literature and it is well established that extreme states of the polar vortex tend to shift Arctic oscillation towards certain states with implications for regional weather conditions. However, the strength of the link between stratospheric conditions and surface weather is poorly estimated because of a low signal-to-noise ratio. To address the low signal-to-noise ratio problem the authors turned their look towards model simulations which provide much more data sufficient to obtain robust estimates of the stratosphere-troposphere coupling. The underlying assumption is that the models provide a reasonable representation of the real atmosphere. The assumption seems to be violated at least in some cases because some estimates obtained from the two different models considered (ECMWF and UKMO) diverge significantly. Which of the two models is closer to the real world is difficult to establish. Therefore, the interpretation of the results should be done carefully. Nevertheless, the article presents novel results which in my opinion go beyond state-of-art. I believe the article can be published in Weather and Climate Dynamics after revision.*

*My main criticism concerns the causal analysis. This should be better described and, in case if the approach used by the authors is a well-established one (I apologize for my ignorance), proper references to background literature should be made. Additionally, I strongly recommend a language check before publication.*

*Major comment:*

*The authors distinguish between increased probability of AO extremes following stratospheric events (their question 2) and how often stratosphere can be considered a cause of extreme AO events (question 3). Both questions are addressed in terms of probabilities (e.g. Figs. 6,7,8). While I understand the difference between the two questions in principle, I do not understand how you manage to solve them separately, given that both questions can only be answered in statistical sense. In figure 7 you show probability of at least one SSW day preceding a randomly sampled day; however in Figure 8 this probability becomes a probability of AO extreme preceded by a SSW day by chance (left panel of Figure 8). This is not the same, clearly. Assume hypothetical world in which all AO extremes are caused by an SSW occurred during previous 30 days. Then dashed lines in Fig. 7 would reach 1 by day -30. However, this would not affect your climatology because it only measures probability of SSW. As a result, you would never be able to correctly answer question 3 using your methodology. For day 30 you would only obtain the difference between 1 and an SSW probability, which is not the right answer to question 3.*

We agree that our method underestimates the true causal relationship, because part of it is masked by the climatological occurrence of SSWs. The effect will be larger if the true causal relation is strong (i.e., almost all AO extremes caused by SSWs) and if the SSWs occur very often in the climatology. What our method hence quantifies the exceedance probability (above climatology) of preceding SSWs to AO– extremes. Nevertheless, this does not represent a rigorous quantification of the causal relation, even under the absence of other processes. We thank the referee for this comment and adapt the manuscript accordingly (see our initial comment for more details).

*Figure 11 illustrates the same problem – there is no evidence in data that AO>3.5 can occur without an SPV within previous 40 days, yet only about half of those events that occurred*

*after SPV can be attributed to SPV following your methodology. I believe the methodology needs to be revised (or I miss something).*

Same issue, the nomenclature and interpretation will be adjusted accordingly.

We reply to a few selected minor comments below. All remaining comments will be addressed in the revised manuscript.

*Other comments*
*L18: What does it mean: "up to a degree of 27%"*
*L31: Please clarify whether you cite daily AO index value, monthly value or seasonally value.*
*L34: Do Kim et al discuss wildfires in winter or in another season?*

Kim et al. report that wildfires occur predominantly around April and find that the annual total burned area in southeastern Siberia is significantly correlated with the average AO in February-March. We will add "in February and March".

*L54: "are needed" for what?*
*L146: Please explain what does "dynamical SSW" mean and provide reference if it has been introduced elsewhere.*

Instead of the commonly used definition of SSWs, which is based on the reversal of u60 from westerly to easterly, e.g., de la Camara et al. (2019) analyze sudden stratospheric deceleration (SSD) events, in order to better account for rapid changes in the dynamics. Our definition of dynamical SSWs forms the intersection of SSWs (accounting, e.g., for fundamentally different wave propagation properties in easterly winds) and SSDs (ensuring a rapid deceleration around the SSW central date). Our results reveal only modest quantitative differences between SSWs and dynamical SSWs, we therefore focus on SSWs only, to allow better comparison with other studies. We will make the description more detailed.

*L151: "we therefore do" what?*
*Figure 1: Although interannual variability of predicted SSW frequency is not the main point of your article I wonder if upper panel of Fig. 1 could show relative frequency of p- SSW rather than absolute numbers. It is quite exciting to see so small number of p-SSWs in 2008/09, a winter in which an SSW occurred in the real world.*

We had decided to show the absolute numbers of p-SSWs so that the contribution of individual years to the overall analysis can be read off. We agree that the relative frequency would also be interesting and we will include a comment about the interannual variability of SSW frequency per winter.

*L165: "the event was generally very rare" sounds strange to me*
*L175: Please provide equation which you apply*
*L197: A rather complicated deseasonalization approach has been used. Why not used a simpler approach in which climatology is estimated using other hindcast years? For example, for ECMWF hindcasts this would provide 19x11=209 realization to build a climatology for each date and lead time. Why do you think it is not enough?*

With our approach, we follow the procedure described here: https://www.ecmwf.int/en/forecasts/documentation-and-support/extended-range/re-forecast-medium-and-extended-forecast-range. As our analyses focus on extreme events, we particularly require an accurate representation of the distributions' tails, which we ensure by including forecasts from within a plus/minus 2 week window. However, this comes at the cost of not accounting for potential leadtime dependent model biases.

*L219: "occur only few days after the event" can you provide the exact lag?*
*L223: I do not think NAM1000 distribution is significantly different from 0 at negative lags.*
*L226: the trend goes to weaker negative values, not positive.*
*L234: I am not sure the name "ECMWF S2S model" is correct.*
*L236: "most phases of negative NAM1000", perhaps: "most cases of negative NAM1000"*
*L243: I do not think NAM1000 in ERA5 follows AR1 process either, or have you checked it?*
*L258: Should not probability of negative NAM be exactly 50%, by construction?*

The NAM index does not follow a perfect Gaussian distribution, therefore, mean and median are not exactly equal.

*Figure 6: What is the period used for calculating the probability increases?*
*L429: I do not think that increasing number of models would help to make definitive quantitative statements unless you know which models are right and which models are wrong. Since all models are different you could only possibly increase the spread.*

We agree and we will describe that including additional models may help to better estimate the robustness of the results.

**Response to comment by anonymous referee #2**

*The authors used a large set of extended-range ensemble forecasts within the sub seasonal-to-seasonal (S2S) framework (namely ECMWF and UKMO models) to obtain an improved characterization of the modulation of AO extremes due to stratosphere- troposphere coupling. Within this framework, they investigated how much stratospheric polar vortex extremes increase the probability of persistently AO phases and their extremes. They found that following potential SSW events, persistently negative AO states (> 1 week duration) are 16% more likely, and the likelihood for extremely negative AO states (< –3σ) is enhanced by at least 35%. How the stratospheric polar vortex extremes can be considered as the cause of the subsequent AO extremes was also quantified and discussed. Despite the straightforward analysis presented in this paper, I still found the results are interesting and the diagnostics can be useful for the forecast model assessment. The main issue I have is a lack of dynamical analysis to explain the differences in the two models in representing the AO extremes followed by stratospheric events (SSWs and SPVs) and the results regarding causal relationships between AO extreme and stratospheric polar vortex extremes. Hence my suggestion is major revisions. Once my points below are answered, I can recommend this work to be published in WCD.*

*General Comments:*
*Two S2S forecast models (ECMWF and UKMO) used in this study showed some quantitative disagreement (i.e., the results diverge significantly e.g., Figs. 5, 6, 7. 8 etc). However, there is no dynamical analysis and explanations to address the issue rather than simply comparing the results in a statistical sense. It would make the results clearer if you could address this issue in the paper.*

We agree that the disagreement between the two models are interesting and that the analysis would benefit from investigating the underlying dynamical causes. We address the observed differences between the models at the beginning of this response.
Beyond that, we think that potential causes are numerous (for instance, differences in wave-mean flow feedbacks or external forcings, e.g., from the tropics). We are in the process of investigating potential sources to the observed differences and we will include the outcome in the discussion section. However, a detailed analysis would go beyond the scope of this study.

*I am not convinced about the causal analysis in this paper. As you are aware, the extreme AO events are not only proceeded by extreme stratospheric events, but also by mid-latitude*

*winter circulation such sea-level pressure, sea-ice and remote forcing from the tropics. How can you isolate the possible stratospheric influence alone from these other factors (since this may not direct/linear statistical relationship)? I believe that not all stratospheric polar vortex extremes lead to AO extremes. You probably need to revise your methodology to address this question.*

We agree that statements about causality in a strict sense are not possible with the kind of analysis presented here. Based on the reviewer's comments we have decided to replace our terminology with more neutral statistical terms such as predictors, probability exceedance and the like. We acknowledge that multiple predictors might exist and not all predictors are based on direct causal relations.

That said, note that other tropospheric drivers of AO events only represent an issue for our statistical inference of stratospheric influence if they have an impact on the stratosphere themselves. Nevertheless, a modification of the stratosphere by such a third driver (e.g., Ural blocking, which may influence the AO directly) might further lead to modifications of the AO. In this sense individual drivers are generally difficult to isolate (because they tend to be coupled), even if their causal connection is clear in principle. Again, we will modify wording to speak of "predictor" in such cases, which should circumvent the issue of common coupled drivers.

As one example of a common quasi-external driver we are in the process of evaluating potential influences of different ENSO states and will include some aspects of these results in our discussion (although a more in-depth study in this direction seems of merit and will be left for future work).

We reply to a few selected minor comments below. All remaining comments will be addressed in the revised manuscript.

*Other Comments:*
*L143: Will be the results sensitive to the WMO's definition that includes the reversal of the meridional temperature gradient?*

We have chosen to define p-SSWs and p-SPVs based on u60 alone, mainly to follow the standard definition of SSWs that is used most often in the literature and to limit the required amount of data storage. Even though we have not explicitly tested, we would expect modest differences in the classification of individual events, but we would expect that differences average out in the composite mean (see, e.g., Butler et al., 2015). Furthermore, our analysis of dynamical SSWs can serve as a sensitivity analysis, and the results show only minor quantitative differences.

*Figure 3. Please also add a similar histogram for UKMO model next to this figure. Also please add the uncertainty in this plot.*
*L175: Please delete this and just mention the number. Otherwise, please provide a full equation before inserting the number.*
*Figure 4. Do you have a similar figure for ERA5? How does it look like compared to UKMO and ECMWF models? It's hard to get definitive quantitative statements since both the model are probably not right.*

We expect ERA5 to be extremely noisy in this diagnostic due to the limited sample size in observational record, but we will check.

*L429: I dont think you will get a definite answer for this rather than a spread of the quantification of the probability of extreme AO events following extreme strat. events in different model configuration.*

We agree, the statement will be modified (see response to RC1).

**Response to comment by editor, Nili Harnik**

*Both reviewers raise concerns about the interpretation in terms of causality, which should be addressed.*

*In addition, both reviewers comment on the need to carefully interpret the results given the large differences between the two models used. In this regard, I am wondering if the following paper, shows biases in the eddy energy spectra in ECMWF IFS model., has anything to do with this difference - Augier and Lindborg, 2013: A New Formulation of the Spectral Energy Budget of the Atmosphere, with Application to Two High-Resolution General Circulation Models. JAS, 2293-2308.*

Thank you, we appreciate the reference to the study. As describe, we are currently investigating which processes could lead to differences in the results and we will consider potential differences in the eddy energy spectra.

---

## Author Response (AR1)

We would like to thank both referees and Nili Harnik again as their constructive comments and suggestions greatly helped us to improve the manuscript. A brief general discussion about the two major concerns raised by the reviewers was provided in our previous comment (AC1, 9th Feb 2022). In the following, we present the specific changes that we have now implemented, accordingly. Our response is divided in three parts: changes based on major concerns, changes based on minor concerns, additional changes. The referees' comments are in *italic gray,* our comments are in blue. Main changes are bold.
* * *
**Changes Based on the referees' major concerns**

1 Description of the methodology and interpretation of causality

Both reviewers raised concerns about the methodology that was applied to quantify the causal relation between polar vortex and subsequent AO extremes.

- We **added subsection 3.5** (Conditional probabilities of polar vortex and AO extremes), where we outline the approach that we follow to quantify the statistical relation between SSWs/ SPVs and AO extremes. Conditioning on stratospheric events allows us to compute the relative probability increase of at least one AO extreme within a subsequent time period. This effectively quantifies the extent to which a stratospheric extreme may act as a predictor of a tropospheric extreme.
  However, quantification of AO extreme occurrence *without* preceding stratospheric extremes, which is needed to assess potential causal links, requires careful evaluation:
  We now make use of the **concepts of attributable risk** (frequently used in other fields such as climate attribution science or epidemiology, see sections 3.5 and 6). In our case the attributable risk **among the exposed** quantifies the fraction of "stratospheric-extreme-exposed" AO extremes that may be attributed to the preceding stratospheric extreme, whereas the attributable risk **among the population** quantifies the fraction of all AO extremes that may be attributed to preceding stratospheric extremes.

- To make the usage of different conditional probabilities clearer, we provide an **overview table** for the applied event definitions (Tab. 2).

- Fig. 7 showed the "estimated probability increase" (AO extremes following SSWs vs. AO climatology). We change the wording to "relative probability increase" to highlight the ratio that appears in the calculation. In addition, we add a plot in the supplement that shows the relative probability increase as a function of the time period that is used to search for AO extremes.

- In section 6, we present composites based on AO extreme events. We compute the probability of negative AO extremes being preceded by SSWs and we had compared this probability to the climatological occurrence of SSWs. We agree with the reviewers that the difference cannot be interpreted as the fraction of AO extremes *being caused* by SSWs (see Fig. 8 of submitted manuscript).
  In the revised manuscript, we **replace Fig. 8** (and Fig. 11., for strong vortex events) with figures that now show that

  - about 50% of $AO^{-3}$ extremes that are preceded by a SSW may statistically be attributed to the preceding SSW, whereas

  - about one quarter of all $AO^{-3}$ extremes may be attributed to preceding SSWs.

- Even though attributable risk is no strict quantification of causality either, it does offer insights into causal links in a statistical sense (see also revised discussion in conclusion section about common drivers).

2 Model Differences

The reviewers noted that given quantitative discrepancies between the models (ECMWF, UKMO) in some of the presented diagnostics, the interpretation needs to be done carefully, and both models could be "wrong". It was suggested that investigating dynamical causes for the observed differences could reveal interesting insights.

- Fig. 6 (relative probability increase of AO extremes following SSWs) shows one of the main results of our study. Considerable discrepancies between the models are observed for negative AO extremes of below –3, as these events occur 40% more often in the ECMWF and 80% more often in the UKMO model, following SSWs. We have now **added the relative probability increase of positive AO extremes following SSWs**, which is negative, i.e., these events become less likely following SSWs. Both models show quantitative agreement for AO thresholds up to about 2.5 standard deviations. To check whether observed differences for thresholds of ±3 stem from sampling uncertainty, we added 95% confidence intervals, obtained via bootstrapping. Indeed, results for AO ≳ ±3 are associated with considerable uncertainties, however, they cannot fully explain the observed differences.
  We believe that potential dynamical sources that can lead to such differences are numerous. They could be related to intrinsic tropospheric dynamics (e.g., related to wave-mean flow feedbacks) or to the manifestation of teleconnections related to external forcings (e.g., from the tropics). Generally, the analysis of extreme events is expected to be very sensitive to even tiny modulations of the underlying distribution. We address the model discrepancies in the discussion.

**Changes Based on the referees' minor concerns**

Referee 1

*Other comments*
*L18: What does it mean: "up to a degree of 27%"*

We have adapted the statement to the new methodology that is used to attribute AO extremes to preceding SSWs. It now says: "3) approximately 50% of extremely negative AO states that follow SSWs may be attributed to the SSW, whereas about one quarter of all extremely negative AO states during winter may be attributed to SSWs."

*L31: Please clarify whether you cite daily AO index value, monthly value or seasonally value.*

We added "daily" [AO index...].

*L34: Do Kim et al discuss wildfires in winter or in another season?*

Kim et al. report that wildfires occur predominantly around April and find that the annual total burned area in southeastern Siberia is significantly correlated with the average AO in February-March. We added "in February and March".

*L54: "are needed" for what?*

We try to make it clearer by adding [Therefore, a very large sample of SSW and SPV events are needed] "to quantify the subsequent risk increase of AO extremes".

*L146: Please explain what does "dynamical SSW" mean and provide reference if it has been introduced elsewhere.*

We expanded the paragraph to explain more detailed how we defined dynamical SSWs as the logical intersection between SSW and SSD (Sudden Stratospheric Deceleration) events.

*L151: "we therefore do" what?*

Thank you for noting, we now write: "[...] we therefore focus on SSWs only, to allow better comparison with other studies."

*Figure 1: Although interannual variability of predicted SSW frequency is not the main point of your article I wonder if upper panel of Fig. 1 could show relative frequency of p- SSW rather than absolute numbers. It is quite exciting to see so small number of p-SSWs in 2008/09, a winter in which an SSW occurred in the real world.*

We appreciate the idea and agree that the relative frequency provides interesting additional information. As we are not aware of a standard procedure to derive seasonal SSW probability from ensemble forecast data, we introduced a **proxy for the seasonal SSW probability**, that is described in detail in **appendix B**. In Fig. 1, we added the proxy as a subplot.

*L165: "the event was generally very rare" sounds strange to me*

We now write that the strong vortex lead to "an only marginal SSW probability in the forecasts", which suggests that "the event itself was unlikely given the prevailing dynamics".

*L175: Please provide equation which you apply*

We decided to only mention the number in the text and refer to the new appendix B. The calculation follows the same procedure as for the newly introduced seasonal SSW proxy.

*L197: A rather complicated deseasonalization approach has been used. Why not used a simpler approach in which climatology is estimated using other hindcast years? For example, for ECMWF hindcasts this would provide 19x11=209 realization to build a climatology for each date and lead time. Why do you think it is not enough?*

With our approach, we follow the procedure described here: https://www.ecmwf.int/en/forecasts/documentation-and-support/extended-range/re-forecast-medium-and-extended-forecast-range. As our analyses focus on extreme events, we particularly require an accurate representation of the distributions' tails, which we ensure by including forecasts from within a plus/minus 2 week window. We added a comment in the conclusions section.

*L219: "occur only few days after the event" can you provide the exact lag?*

We added "(lag +2 days: $-6ms^{-1}$)".

*L223: I do not think NAM1000 distribution is significantly different from 0 at negative lags.*

We have tested whether the described positive anomalies are significantly different from 0 via a two-sided student's t-test and find p-values below $10^{-10}$ in both ECMWF and UKMO models (as a result of the large sample size). However, we agree that physical interpretation requires further research.

*L226: the trend goes to weaker negative values, not positive.*

Corrected, thank you.

*L234: I am not sure the name "ECMWF S2S model" is correct.*

We now write "ECMWF model".

*L236: "most phases of negative NAM1000", perhaps: "most cases of negative NAM1000"*

Adapted as suggested, thank you.

*L243: I do not think NAM1000 in ERA5 follows AR1 process either, or have you checked it?*

Based on the concerns raised, we have checked again and identified a bug in the previous calculation for AO persistency in ERA5 (thank you!). It turns out that ERA5 and ECMWF (S2S) agree very well in their climatologies, suggesting that the observed variability cannot be reproduced by an AR1 process. We have updated the text accordingly.

*L258: Should not probability of negative NAM be exactly 50%, by construction?*

The NAM index distribution turns out to be not perfectly Gaussian, therefore, mean and median are not exactly equal. We now write: "Asymmetry between positive and negative values arises from the AO distribution that is not perfectly Gaussian (skewness: -0.13)."

*Figure 6: What is the period used for calculating the probability increases?*

We added that the we compute the relative probability increase by averaging over periods of 25 days to 40 days.

*L429: I do not think that increasing number of models would help to make definitive quantitative statements unless you know which models are right and which models are wrong. Since all models are different you could only possibly increase the spread.*

Yes, we agree and we removed the sentence. We provide a brief discussion about observed discrepancies between ECMWF and UKMO. Including further models would likely raise similar issues.

Referee 2

*L143: Will be the results sensitive to the WMO's definition that includes the reversal of the meridional temperature gradient?*

We have chosen to define p-SSWs and p-SPVs based on u60 alone, mainly to follow the standard definition of SSWs that is used most often in the literature and to limit the required amount of data storage. Even though we have not explicitly tested, we would expect modest differences in the classification of individual events, but we would expect that differences average out in the composite mean (see, e.g., Butler et al., 2015). Furthermore, our analysis of dynamical SSWs can serve as a sensitivity analysis, and the results show only minor quantitative differences.

*Figure 3. Please also add a similar histogram for UKMO model next to this figure. Also please add the uncertainty in this plot.*

Both histograms for ECMWF and UKMO agree extremely well. UKMO forecasts show very long periods (>28 days) of persistently negative AO slightly more often, which is likely due to the longer leadtime. In addition, sampling uncertainties are very small, such that error bars denoting the 95% confidence intervals are even hard to see. Therefore, we decided to omit error bars as well as the histogram for UKMO data, but we added a comment.

[Figure]

[Figure]

*L175: Please delete this and just mention the number. Otherwise, please provide a full equation before inserting the number.*

We now provide only the number and refer to the new appendix B for further details.

*Figure 4. Do you have a similar figure for ERA5? How does it look like compared to UKMO and ECMWF models? It's hard to get definitive quantitative statements since both the model are probably not right.*

We have checked: AO<0 is more likely in ERA5 at positive lags (between 55% and 75%) and even at negative lags already (up to 60%). However, 95% confidence intervals span up to 40% and therefore reveal large associated sampling uncertainties. Neither AO<–3 nor AO>+3 events are observed following SSWs in ERA5, likely due to the limited sample size. We added a comment.

*L429: I dont think you will get a definite answer for this rather than a spread of the quantification of the probability of extreme AO events following extreme strat. events in different model configuration.*

Yes, we agree and we removed the sentence. We provide a brief discussion about observed discrepancies between ECMWF and UKMO. Including further models would likely raise similar issues. [see response to referee 1]

**Additional Changes**

- To avoid confusion and to allow for shorter abbreviations for event definitions, we changed the terminology from NAM1000 (Northern Annular Mode at 1000hPa) to AO (Arctic Oscillation).

- We use the notation $AO^-$ for negative and $AO^+$ for positive AO events. Particular thresholds are explicitly indicated, e.g., $AO^{-3}$.

- We adjusted the way we compute the probabilities for "at least one event within time t", e.g. $P(AO_{wt} \mid SSW)$. Before, we had computed daily event probabilities and derived "at least one" by: 1 minus "no event on every day". Now, we compute the probabilities by explicitly checking how many of the available forecasts fulfill the respective conditions. Both approaches are expected to yield the same result in the limit of large forecast sample sizes.

---

## Author Response (AR2)

Many thanks again to both referees for their comments. We are happy that referee 2 has accepted our manuscript for publication. Below, we address the comments that were given to us by referee 1. The referee's comments are in *italic gray*, our responses are in blue.
(Line numbers refer to the updated manuscript.)

*This is my second review of the manuscript by Spaeth and Birner. The authors have addressed most of my minor comments; however, the comment regarding attribution analysis has not been properly addressed. In the revised version the authors introduce the FAR concept and apply it to attribute AO events to SSW (or SPV) events. I believe this is just another way to quantify the changed probability of AO events following SSWs, similar to relative probability increase (RPI). Furthermore, FAR analysis leads to confusing conclusions such as "approximately 50% of extremely negative AO states that follow SSWs may be attributed to the SSW". To make such statements one needs to block the SSW-AO link in these same SSW situations and see how many AOs would occur in such controlled experiments. This has not been done. Instead, what FAR indeed shows is the increased probability of AO following SSW, similarly to RPI. The difference between FAR (50% increase) and RPI (35-40% increase) is likely because the authors compare FAR to no-SSW situation while in RPI they compare SSW cases to all AO occurrence probability. Further, what the authors call "FAR among the population" looks like a decreased probability of AO during no SSW periods. In summary, I recommend the authors to modify the language and clarify that what they are doing are different ways to estimate increased probability of AO due to SSW occurrence. Without this I cannot recommend the manuscript for publication.*

We appreciate the reviewer's concern about what can and cannot be inferred from the kind of attribution analysis we have done. Nevertheless, we are convinced that useful insights may be obtained by applying an FAR framework to SSW-AO coupling that go beyond of what simple probability increase measures provide.

First, let us summarize where we have adapted the manuscript in line with our arguments (see below):
- A paragraph was added at the end of the introduction that is aimed to clarify what is meant by "attributable to" (ll. 85ff): In particular, it is outlined that RPI and FAR may capture pathways other than the direct SSW-AO link.
- We are now more careful with terminology that would suggest causal relations (e.g., ll. 81-84, 393, 410ff, 416, 430, 441, 447, 479). Our first two research questions now read "By how much is the probability [...] increased following stratospheric polar vortex extremes?" (ll. 81ff; instead of: "By how much do stratospheric polar vortex extremes increase [...]"). In the abstract and in the answer of the third research question, we added a caveat concerning limited causal validity (ll. 20f, 518f). In the previous manuscript, we had summarized the results of FAR using the wording "may be attributed to". In the revised manuscript, we adjust the expression to "may be attributable to" in order to stress that attribution is only possible under certain assumptions. We chose not to use a weaker expression like "associated to" as this might suggest (the incorrect) interpretation to include fortuitous co-occurrences, which are indeed ruled out by our statistical estimates. Furthermore, the wording "may be attributable to" is terminologically consistent with FAR, therefore it implies on which analyses the conclusions are based on.
To account for the restrictions that arise from a finite sample size and potential biases in the S2S models, we replaced verbs like "determine" and "quantify" by "estimate" (e.g., ll. 96, 233, 294, 365, 434).
- In section 3.5, we clarified that RPI and FAR are based solely on (conditional) probabilities and aim to quantify enhanced AO-occurrence following SSWs (ll. 79, 262ff). The explanations of RPI and FAR are slightly extended (ll. 251ff, 259).
- We extended the discussion of our results in the context of causality (ll. 534ff, 564ff). We explain that our results do themselves not allow strict causal statements. However, existing knowledge about pathways, involving the direct link between stratospheric and tropospheric extremes, but also teleconnections that act as common drivers, help to interpret RPI and FAR.

- further minor changes (see lines given in the discussions below and see differences file created via *latex-diff*)

Concerning the question of different statistical measures (RPI, FAR), we agree that they quantify, in different ways, the fact that AO- events occur more often following SSWs than without (see new ll. 79, 262ff). Our RPI definition does not include information about the "unexposed" part of the population, which is not well defined when applying a "forward" analysis: given SSW happened, what is increased likelihood for AO within certain time period following SSW (the only meaningful reference likelihood is base rate of AO occurrence, see ll. 242ff, 380ff, 396ff in manuscript). For our FAR estimates we take advantage of a "backward" analysis, where we change the time period in the event definitions: e.g., given SSW happened within certain time period, what is the likelihood for AO on the day following that period. This allows estimation of "unexposed" probabilities.

We are convinced that our FAR definition does provide a true estimate of the SSW effect on AO occurrence (=attributable risk), provided the following assumptions are met: 1) sufficiently large sample size, 2) unbiased S2S model, 3) common drivers can be ruled out. We believe that assumption 1) is certainly fulfilled and assumptions 2) and 3) are, although not completely, sufficiently fulfilled to the extent that a first estimate of FAR in the context of strat-trop coupling is feasible (as far as we're aware no previous such estimate exists).

We hope that future studies will be able to improve these estimates, in particular via addressing assumption 3). Uncertainties that arise from the influence of common drivers could be reduced by systematic conditioning (Kretschmer et al., 2021). Furthermore, "intervention experiments" can serve to isolate the SSW-AO pathway, which we indeed plan to perform and exploit ourselves.

Note that real world, "controlled" experiments would be needed to infer strict causality, but this is, of course, not possible. The same situation applies in epidemiology, where FAR was initially introduced. There the effect of exposure is impossible to assess from the fraction of population that has been exposed ("controlled experiments" are impossible as well). Instead an unexposed group is used to estimate the effect. FAR simply serves as a useful statistical measure to obtain an estimate of the effect. It turns simple conditional probabilities into more informative risk measures. Nevertheless, the usual caveats about any inferred causality apply (unaccounted-for common drivers, impossibility to "intervene" etc.) and we have revised the text further to be more clear about them.

Regarding comparison of our RPI and FAR, it is important to note that they refer to different event definitions. As pointed out by the reviewer, RPI compares AO occurrence following SSWs to climatological AO occurrence. FAR requires an "unexposed" AO likelihood, which we obtain by switching the definitions "SSW on day 0" to "at least one SSW within a time period" and "at least one AO within a time period" to "AO on the day after the (no-) SSW period". Therefore, the applied event definitions are themselves different, regardless of whether comparing AO occurrence to "climatology" (as in RPI) or to "without SSW" (as in FAR).
Second, and more importantly, we argue that our RPI and FAR aim to answer different questions, despite the fact that the two measures are indeed depended (because both rely on the increased likelihood of AO extremes following certain stratospheric events).
RPI quantifies the increased risk for an AO event following a SSW relative to its climatological occurrence. Hence, it ranges between -1 and $1/P(AO) - 1$.
FAR is itself not a risk, but quantifies the fraction of events in addition to fortuitous SSW-AO co-occurrence. This fraction can take values from 0 to 1. Furthermore, $FAR_p$ incorporates the SSW (=exposure) frequency, which is not the case for RPI and $FAR_e$.
Because of these differences in their definitions, a one-to-one relation between RPI and FAR is not possible and both measures require different interpretations (see, e.g., ll. 564ff).

*Minor comments:*
*L11: expand ECMWF abbreviation*

adapted.

*L35-36: Add in "April"*

adapted.

*L169: change "quasi-deterministic forecast range of ten days" to "first ten days"*

adapted.

*L280: "process, indicating an AR1 process cannot reproduce the observed AO variability"*
*This statement does not make sense to me. You compare duration of AO in ECMWF with an*
*AR1 process inferred from autocorrelation function based on observations.*
*How can you conclude from this comparison that AR1 cannot reproduce observed duration?*

We added that the AO lag-1 autocorrelation agrees well between ERA5 (0.91) and ECMWF forecasts (0.88). The climatological occurrence of persistent negative AO phases is also very similar in the two datasets. The AR1 process shows, however, significantly more often short periods and less often long periods of negative AO, compared to both ERA5 reanalysis and ECMWF forecasts. We now write:

"[...], indicating an AR1 process cannot reproduce AO variability." (l. 206)

*L313-315: The observed probability of AO-, given SSW, is not reported, but the confidence*
*intervals are provided. Why not report the observed probability?*

The observed probability for AO− given SSW fluctuates strongly with leadtime and is between ~0.55 and ~0.8, depending on the lag. The corresponding confidence interval is apparently also a function of lagtime. Due to the significant fluctuations, we do not want to draw too much attention on the reanalysis and therefore only provide a range of possible probabilities in terms of an overall estimated confidence interval.

*L316: The observed baseline probability of AO-3sigma is not reported, what does it mean*
*"modestly lower"?*

We now provide the exact AO extreme base probabilities for ERA5: $AO^{-3}$ = 0.06%; $AO^{+3}$ = 0.02%. We added that the lower extreme probabilities in ERA5 are consistent with a negative kurtosis of the ERA5 AO distribution.

*L345: Should not you multiply relative probability increase by 100% if you show it in Fig. 6 in*
*percentages?*

We added in the subtitle of Fig. 6 that the probability increase is shown in percent.

*L348: Should not relative probability increase approach 0 in the limit of large t?*

Yes, the limit of 1 referred to the fraction. To clarify, we changed to:

"The ratio is a function of the length of the time window t (see supplement Fig. S2). In the limit of large t, where the SSW influence becomes negligible, it is expected to approach 1, such that the relative probability increase approaches 0."

*L392: In the denominator: P(noSSWwt)=1-P(SSWwt)*

adapted, thank you for spotting!

*L449: Why only these two equations are numbered? Please number all equations.*

adapted.

*L478: the abstract report at least 35% increased AO- probability after SSW. Here, the number is 40%. You should be consistent.*

adapted.

*L504-505: Yes, causality in the stratosphere-troposphere coupling works in both directions, however, an AO event that occurs after an SSW can't cause the latter.*

We have slightly modified the paragraph. The sentence was moved to a footnote and we added a note:

"It is important to keep in mind that the coupling is, in general, mutual and causality works in both directions (even though given events can of course not affect the past)."

*L630: Allen et al. (2003) does not discuss FAR and is irrelevant to your study. The reference you need is: Allen M.R. (2003). Liability for climate change. Nature 2003, 421:892.*

adapted, thank you!
* * *
Kretschmer, M., Adams, S. V., Arribas, A., Prudden, R., Robinson, N., Saggioro, E., & Shepherd, T. G. (2021). Quantifying Causal Pathways of Teleconnections, Bulletin of the American Meteorological Society, 102(12), E2247-E2263. Retrieved Jun 15, 2022, from https://journals.ametsoc.org/view/journals/bams/102/12/BAMS-D-20-0117.1.xml